# Experimental Study on Triaxial Compressive Mechanical Properties of Polypropylene Fiber Coral Seawater Concrete

**DOI:** 10.3390/ma15124234

**Published:** 2022-06-15

**Authors:** Hang Shi, Linlin Mo, Mingyan Pan, Leiguo Liu, Zongping Chen

**Affiliations:** 1College of Civil Engineering and Architecture, Guangxi University, Nanning 530004, China; hangshi818@163.com (H.S.); linlinmo6688@163.com (L.M.); p1037413821@163.com (M.P.); llg1206782407@163.com (L.L.); 2College of Architecture and Civil Engineering, Nanning University, Nanning 530200, China; 3Key Laboratory of Disaster Prevention and Structure Safety of the Ministry of Education, Guangxi University, Nanning 530004, China

**Keywords:** polypropylene fiber, all-coral seawater concrete, triaxially pressurized, confining pressure value, fiber dose, lightweight structural concrete

## Abstract

In order to study the mechanical properties of polypropylene fiber all-coral seawater concrete in triaxial compression, 36 specimens were developed and constructed for triaxial compression load testing employing confining pressure value (0, 6, 12, 18 MPa) and polypropylene fiber admixture (1 kg·m^−3^, 2 kg·m^−3^, 3 kg·m^−3^) as variation parameters. The test observed the failure mode of the specimen and obtained the stress–strain curve of the whole process of its force damage failure. An in-depth analysis of polypropylene fiber all-coral seawater concrete’s peak stress, peak strain, initial elastic modulus, axial deflection, energy dissipation, ductility, and damage evolution process was carried out based on the experimental data. The test findings indicated that the best effect on the deformation properties of polypropylene fiber all-coral seawater concrete is obtained when 3 kg·m^−3^ of polypropylene fiber is blended. Under triaxial compression, the correct number of polypropylene fibers may significantly enhance the peak stress, peak strain, ductility, and elastic modulus of polypropylene fiber all-coral seawater concrete, therefore enhancing the brittle characteristics of coral concrete. During the triaxial surround pressure test, the confining pressure value and polypropylene fiber coupling effect delayed the appearance of initial damage in polypropylene fiber complete coral seawater concrete specimens, slowed the development of damage, and reduced the degree of damage to the specimens.

## 1. Introduction

In recent years, with the continuous depletion of land resources, the demand for the construction of oceanic islands and reefs has continued to increase. The construction of islands and reefs not only involves the issue of territorial integrity and sovereignty of countries, but also has significant importance in military and economic terms. However, traditional gravel aggregates and freshwater resources on the island are extremely limited, and transportation from the mainland is costly and time-consuming [1]. Coral debris is light, porous, highly absorbent, loose in structure, rough on the surface, and is a natural lightweight aggregate [2]. Coral debris is used as coarse aggregate. Coral sand is used as fine aggregate, and the concrete mixed with seawater and cement is all-coral seawater concrete. Applying all-coral seawater concrete as a green and environmentally friendly building material has important scientific significance and practical engineering value in the engineering fields of islands, reefs, breakwaters, roads, and airports in coastal areas due to its advantages of light weight and easy access to materials. However, coral inevitably has local cracks and infiltration as a porous material. Given the congenital disability that is prone to micro-cracks, if carbon steel is utilized, the corrosion resistance is poor in the harsh marine environment [3,4]. The addition of uniformly dispersed short fibers can significantly improve the defects of poor toughness and easy cracking of plain concrete [5], and microfibers can delay the development of macroscopic cracks [6]. Polypropylene fibers thicken concrete, reduce concrete slump, improve crack resistance performance, and inhibit early shrinkage cracks in concrete [7]. Given this, polypropylene fibers (PPF) with good toughening and fracture-resisting capacity, corrosion resistance, odorlessness, nontoxicity, high toughness, low price, and good chemical stability were chosen to be effectively dispersed in concrete [8].

So far, scholars from various countries have achieved more fruitful results on the mechanical properties of coral concrete [9,10,11]. In 1974, Howdyshell P A [12], a U.S. Army Construction Engineering Laboratory scholar, published a research study on coral concrete and concluded that “it is feasible to formulate concrete using coarse coral aggregates, but the chloride salt content of coral aggregates should be controlled”. Huang et al. showed through experimental comparison analysis that seawater increased the strength and elastic modulus of coral concrete compared to freshwater [13], and in 1991, Rick et al. [14] examined three coral concrete buildings on Bikini Island in the Pacific Ocean. They concluded that “the strength of coral concrete can meet the design requirements of engineering structures”. In 1996, the Indian scholar Arumugam, R.A. et al. [15] conducted an extensive study and confirmed that coral concrete has a high early strength growth and a slow late strength growth. Daboo et al. [16] studied the mechanical properties of all-coral seawater concrete during uniaxial compression and showed that the uniaxially compressed specimens of all-coral seawater concrete still have high residual strength after damage and have some ductility. Zhang et al. [17] used fiber-reinforced polymer to improve seawater coral aggregate concrete structures’ bearing capacity and service performance in maritime environments. The findings indicate that the specimen has a significant mechanical bite force at the interface, enhances the mechanical interaction of the slurry–aggregate contact, and increases the specimen’s bond strength. Wang et al. [18] conceived the unidirectional axial compression test of an FRP steel composite circular tubular concrete column with circular saltwater, sea sand, and coral aggregate. According to the findings, the maximum stress and strain experienced by the specimen experience a discernible increase when the number of FRP layers in the structure is increased. The experimental consequences of Arefi et al. [19] showed that coarse polypropylene fibers improved the ultimate load-bearing capacity and energy absorption capacity of concrete. Bagherzadeh et al. [8] concluded that the polypropylene fibers (PPF) with a length of 19 mm improved the toughness index of concrete. Wang et al. [20] pointed out that polypropylene fibers make concrete with lower elastic modulus, reduced shrinkage, and more robust deformation performance with the same water–cement ratio and the same slump conditions. Xu et al. [21] concluded from uniaxial cyclic loading tests that the addition of polypropylene fibers can significantly improve the mechanical compressive cyclic behavior of concrete. It increased the compressive toughness and ultimate ductile peak hysteresis energy capacity of concrete. The concrete’s stiffness degradation and stress deterioration were reduced. The subtle blending rate of synthetic fibers can play a significant role in concrete crack resistance and toughening [22]. Too-high fiber blending significantly reduces its workability [23].

From the above research results, it can be seen that there has been much research on the basic mechanics of polypropylene-fiber-reinforced concrete. However, there are very few studies on polypropylene-fiber-reinforced all-coral seawater concrete. Especially in engineering practice, most triaxial-force-bearing components–such as revetment, retaining walls, and other port engineering structures and wave-absorbing blocks–are subjected to complex stress states. Scholars have paid very little attention to its mechanical properties and damage and energy consumption processes. Therefore, this paper intends to carry out triaxial compressive tests of polypropylene fiber all-coral seawater concrete and carry out the ratio design of all-coral seawater concrete with different doses of polypropylene fiber. The effects of peritectic pressure value and fiber content on the peak stress, peak strain, elastic modulus, ductility, actual energy consumption, and damage development process of polypropylene fiber all-coral seawater concrete were analyzed in depth. Also put forward were the force mechanism and the functional relationship of each design parameter. At present, the research results of coral aggregate concrete basically focus on the strength level, or uniaxial compression test, and there is no in-depth discussion on the triaxial stress mechanism and stress performance of coral coarse aggregate concrete. This experiment was carried out for the first time in the world and has certain research significance. In this paper, the modifying effect of polypropylene fibers on the mechanical properties of all-coral seawater concrete under complex stress conditions is investigated. In order to provide reference for applications such as marine engineering island construction.

## 2. Experimental Overview

### 2.1. Test Materials

The test materials included coarse coral aggregate, fine coral aggregate, seawater, ordinary Portland cement, polypropylene fiber, and water-reducing agent. Among them, coral aggregate is selected as coarse aggregate and coral sand as fine aggregate on an island in the South China Sea (location: 21°27′28″ N, 109°22′36″ E), as shown in Figure 1. According to “Light Aggregate and its test method” (GB/T 17431.2-2010) [24], the primary physical property indexes of coral aggregate were tested, and the results are shown in Table 1. The fibers were selected from polypropylene fibers (PPF), as shown in Figure 2. Its main performance indexes are shown in Table 2. The seawater in the mixing water comes from the original seawater of a sea area in the South China Sea, and its main components are shown in Table 3. The cement adopted Conch P-O42.5R ordinary Portland cement, and the water-reducing agent adopted polycarboxylic acid water-reducing agent.

### 2.2. Test Piece Design

The ratio design is shown in Table 4. According to the polypropylene fiber parameters suggested by the Technical Specification for Fibrous Concrete Structures (CECS38:2004), polypropylene fiber blending amounts of 1 kg·m^−3^, 2 kg·m^−3^, and 3 kg·m^−3^ were selected. Since the test was conducted using coral aggregates that were not prewetted, the feeding sequence shown in Figure 3 was used in mixing the concrete. After the specimens were made and demolded, they were loaded after 28 days of maintenance at a temperature of 20 ± 2 °C and a relative humidity of greater than 95%. Twelve groups (three per group) of cylindrical specimens with diameter *D* = 100 mm and height *H* = 200 mm were designed and fabricated with the various parameters of confining pressure value *σ_w_* and polypropylene fiber dosing *V*. The ratio design of polypropylene fiber all-coral seawater concrete was referred to the “light aggregate concrete structure technical regulations” (JGJ 12-2006) [25]. The design strength grade of concrete is C30. The specific design parameters of each specimen are shown in Table 5.

### 2.3. Loading Device and Loading Method

All specimens were loaded using an RMT-201 rock mechanics and concrete mechanics testing machine following the specifications of Mechanical Properties Test Methods for Ordinary Concrete (GB/T 50081-2019) [26], and the specimens were placed in a triaxial pressure chamber. The axial electrohydraulic servo actuator and the combining pressure pump jointly applied triaxial pressure to the specimens, and the loading device is shown in Figure 4.

When the specimen was loaded, it was initially prepressurized to the hydrostatic pressure condition shown in Figure 5, at which point the specimen’s lateral pressure equaled the planned target confining pressure value. Next, the displacement-controlled loading system was used with a loading rate of 0.02 mm/s under the condition that the confining pressure value was kept constant. As shown in Figure 6, the loading system’s turning point is when the axial load on the specimen drops to 85% of the peak load or when the axial deformation is inconveniently loaded.

Gemini SEM 360 was used for microscopic observation, as shown in Figure 7. The images of polypropylene fiber, coral aggregate, coral aggregate slurry interface transition zone, fiber aggregate interface transition zone, and fiber slurry interface transition zone were produced at varying confining pressures.

## 3. Experimental Results and Analysis

### 3.1. Specimen Damage Analysis

Figure 8 shows the damaged specimens. Using PCAS software to map the specimen’s fractures, it can be seen that the damaged specimen changed from axial splitting failure to diagonal shear failure and finally to transverse shear failure [27].

When the confining pressure value *σ_w_* = 0 MPa, the specimen was in a uniaxial compression state, one or more main cracks parallel to the loading direction existed on the surface at the time of damage, and the specimen was damaged by vertical splitting at the end of loading, as shown in Figure 8a,e,i. When the constraint pressure value was 0 MPa ≤ *σ_w_* ≤ 6 MPa, the vertical development of cracks inside the whole coral concrete was restricted due to the constraint effect of the constraint pressure value. The main crack developed obliquely, and the specimen showed oblique splitting damage. The horizontal angle of the main crack was approximately 65°. With the increase in the restraint pressure value, the angle slightly decreased, and the width of the main crack gradually decreased. When the confining pressure reached 6 MPa ≤ *σ_w_* ≤ 12 MPa, the horizontal angle of the main crack continued to shrink to between 45° and 60°. At this time, due to the large deformation of the specimen, the confinement effect of the confining pressure value on the fine cracks began to weaken. It was observed that the specimen exhibited oblique shear failure, the oblique crack penetrated the specimen, and the crack developed from oblique to approximately horizontal, forming many horizontal cracks. The failure surface was primarily concentrated in the middle of the specimen. When the confining pressure value was 12 MPa ≤ *σ_w_* ≤ 18 MPa, the higher lateral restraint dramatically limited the development and penetration of vertical and oblique cracks, and the failure mode changed to transverse shear failure. There were no apparent cracks on the surface, the oblique prominent cracks on the surface of the specimen largely disappeared, and the grid-like cracks gradually increased. The stress peak point disappeared, and the peak of the curve was nearly horizontal. At the end of loading, the middle or end of the specimens with more dense grid-like cracks became thicker.

The effect of polypropylene fiber on the damaged specimens was more evident when the confining pressure value was lower than 6 MPa, and the effect of polypropylene fiber on the destructive form gradually decreased as the confining pressure value increased. Among them, the minimum number of visible cracks on the surface of the specimen appeared with polypropylene fiber content *V* = 3 kg·m^−3^. During the test, at *σ_w_* = 0 MPa, the specimens that did not contain polypropylene fiber sustained severe, broad cracking and spalling damage. Some specimens even fragmented into many pieces. The sample containing polypropylene fiber broke but did not peel off. When the specimen’s compressive strength was attained, two symmetrical diagonal fractures occurred on the specimen’s surface for the whole coral concrete with fiber admixture *V* = 1 kg·m^−3^ and confining pressure value *σ_w_* = 0 MPa. In consequence, the specimen lost its axial-load-bearing ability and failed.

### 3.2. Damage Interface Characterization

Figure 9 presents the fiber and aggregate distribution in the longitudinal section of the specimen, which shows that the fibers were uniformly mixed between the coarse coral aggregate and coral sand, and the coarse coral aggregate and coral sand were uniformly distributed with few pores. Fibers still played a bridging effect after the concrete was destroyed. As shown in Figure 10, due to the bridging effect of fibers, the formation and development of cracks in the matrix were inhibited, the degree of damage to the matrix was reduced, and its resistance to deformation was improved. After reaching the maximum load, it still did not break, which significantly reduces the possibility of material failure. In the center of the damage, the concrete did not induce spalling but rather drum-like expansion. It demonstrates that polypropylene fiber all-coral seawater concrete retains a specified load-bearing capability after destruction and that the crack width lowers dramatically as the fiber additive content rises. The disordered distribution of polypropylene fibers at different scales hinders the multiplication and expansion of cracks in the matrix. The two bodies broken by shear cracking in the experiment were firmly connected due to the linking effect of the fibers.

### 3.3. Stress–Strain Curve

Figure 11 shows the stress–strain curves of each group of specimens. From the figure analysis, it can be seen that the the confining pressure value had a significant effect on the properties of the specimens, such as peak stress, peak strain, and elastic modulus. When *σ_w_* = 0 MPa, the specimen was in uniaxial compression. Its stress–strain curve experienced a rapid decrease in strength after the peak. This indicates that the all-coral seawater concrete has prominent brittle damage characteristics in the uniaxial compression state. Based on the analysis of 0MPa ≤ *σ_w_* ≤ 6 MPa, the slope of the ascending section of the stress–strain curve of the specimen increased with the increase in the restraint pressure value. The peak rose step by step, and the descending section gradually became flat. It indicates that a certain lateral restraint can block the interfacial bond damage and ensure the load-bearing and deformation capacity of polypropylene all-coral seawater concrete. When 6 MPa ≤ *σ_w_* ≤ 12 MPa, the peak of the stress–strain curve continued to increase, the peak of the curve tended to be horizontal, and the damage rate slowed down. The residual deformation of most specimens under this circumferential pressure was considerable, and they were undamaged after loading. When *σ_w_* = 18 MPa, the peak point of the stress–strain curve disappeared, and the curve continued to rise slowly after reaching the peak value. At the same time, a sizeable axial deformation occurred, indicating that the strength and deformation performance of the all-coral seawater concrete had been improved under the constraint of high confining pressure.

The peak stress decreased and then increased, and the peak strain increased and then decreased during the increase in polypropylene fiber addition from *V* = 1 kg·m^−3^ to *V* = 3 kg·m^−3^.

After the stress–strain curve experienced peak load, the strength of the specimen with fiber admixture *V* = 0 kg·m^−3^ decreased rapidly. When the fiber content increased from *V* = 1 kg·m^−3^ to *V* = 2 kg·m^−3^, the polypropylene fibers started to participate in stopping cracking in the descending phase of the stress–strain curve as the cracks inside the specimen gradually expanded. The decreasing section of the stress–strain curve of the specimen gradually leveled off. The residual strength gradually increased, making the development of fine cracks in the cracking of all-coral seawater concrete delayed. When the fiber content was *V* = 2 kg·m^−3^ to *V* = 3 kg·m^−3^ due to the correct amount of polypropylene fiber in the full coral seawater concrete, specimens under load inhibited the development of small cracks to a certain extent. A high number of fibers in a certain amount of coral concrete could not be thoroughly dispersed. Clumped and knotted fibers formed weak interfaces in the all-coral seawater concrete. Cement slurry could not enter into the concrete, creating an internal stress concentration damage point, thus affecting the compressive strength.

### 3.4. Curve Feature Point Parameters

Each group of specimens had characteristic point parameters such as yield, peak, stress–strain parameters at the damage point, and elastic modulus (secant modulus at 0.4 *σ_v_* in the rising section of the stress–strain curve), as reported in Table 6. The PF-2-18-C specimen was eliminated due to the large dispersion of the test results.

### 3.5. SEM Images and Analysis

The structure of polypropylene fibers was observed by scanning electron microscopy (SEM), and the surface of polypropylene fibers was found to be smooth and in the form of long thin strips, as shown in Figure 12. Figure 13 is the SEM image of the coral coarse aggregate. It was found that the surface of the coral aggregate was rough and porous, and a large number of layered and cage-like structures were distributed inside. There are many tiny crystals on the surface of coral sand particles, which are highly irregular, have ups and downs, and have high particle roughness. Coral aggregates have a more pronounced apparent density and internal friction angle relative to natural aggregates. The internal friction angle helped to increase the interfacial bond between the aggregates and the hardened mortar. To some extent, it improved the interfacial strength of concrete [28].

Figure 14 shows the coral aggregate–cement paste interface transition zone. The surface of the coral aggregate is densely covered with micropores into which the cement paste penetrated. Therefore, this type of micropore enhances the mechanical engagement between aggregate and cement stone. The high water absorption of coral aggregate avoided the delamination between aggregate and cement caused by water enrichment. The increase in the surrounding pressure value made the cement stone more compact. The large amount of hydration products formed also filled the voids at the interface, making the bond between the aggregate and cement paste tighter.

Figure 15 and Figure 16 are the fiber–cement paste interface transition zone and the fiber–aggregate interface transition zone. The incorporation of polypropylene fiber reduced the porosity and macropore volume of the interface transition zone, making the internal structure denser and having a beneficial effect on the strength of the concrete. Fibers are equivalent to fine bars, and their three-dimensional random directions were evenly distributed in the concrete to play an anchoring role. They prevented aggregate segregation and reduced the presence of the water-gathering film on the aggregate surface, thereby reducing bleeding pores and improving the microstructure of the transition zone between cement and coral aggregates. As the confining pressure value increased, the lateral deformation of the coral aggregate was restricted. The overall integrity and stability of the specimen improved. The coral aggregate–cement paste interface and fiber–coral aggregate interface were more closely connected. The smaller the crack width, the fewer pores and the more pronounced the squeezed shape of the slurry.

## 4. Influencing Factors and Analysis

### 4.1. Peak Stress

All specimens’ peak stress and confining pressure values were normalized to obtain the variation relationship between the confining pressure value and the peak stress shown in Figure 17. With the increase in the confining pressure value, the peak stress of polypropylene full coral seawater concrete was significantly enhanced, showing a nonlinear increment, and the growth rate of the peak stress gradually decreased with the increase in the confining pressure value. The peak stress of the specimen at the confining pressure value of 18 MPa was 4.66 times that in the uniaxial compression case. The lateral restraint effect of the confining pressure value reduced the risk of local instability in the concrete. The development of microcracks was limited, the generation of vertical penetration cracks was avoided, and the axial bearing capacity of the specimens was significantly improved.

Firstly, the specimen’s lateral deformation was restricted by the constraining impact of the confining pressure value, leading to an increase in the axial load capacity and a rise in peak stress. Increasing the value of the confining pressure resulted in a sharp increase in the deformation capacity of the concrete. The part beyond the material’s own deformation was compensated by the development of cracks. The fine cracks inside the specimen steadily expanded. Coral aggregate has a certain brittleness, so the higher the load that the specimen is subjected to when it is under hydrostatic pressure, the easier it is for the specimen to produce microcracks that affect the peak stress before the effective loading. Large confining pressure values and axial pressures can easily lead to brittle fracture or crushing of coral aggregates, resulting in a slower growth rate of peak stress.

According to the test data, we normalized the peak stress of each group of specimens to combine with the confining pressure value for dimensionless analysis and fit to obtain the formula for calculating the peak stress of polypropylene fiber all-coral seawater concrete under different confining pressure values:(1)$$\frac{\sigma_{v}}{\sigma_{0}}=3.6{\left(\frac{\sigma_{w}}{\sigma_{0}}\right)}^{0.87}+0.97,\;R^{2}=0.97$$

*σ_0_* denotes the peak stress of the specimen under a uniaxial compression state; *σ_v_* denotes the peak stress of each group of specimens under different circumferential pressures; *σ_w_* denotes each group corresponding to the confining pressure value.

Figure 18a shows the peak stress variation relationship for specimens with different polypropylene fiber contents. As shown in the figure, the peak stress was highest for fiber doping *V* = 3 kg·m^−3^ and lowest for fiber content *V* = 2 kg·m^−3^ at a low confining pressure value. With the increase in the confining pressure value, the peak stress of the specimens with *V* = 1 kg·m^−3^ fiber addition was higher than that of other ratios, and the effect of different fiber doping was gradually reduced.

Figure 18b shows the variation of polypropylene fiber content concerning the average peak stress of the specimens under each circumferential pressure. As seen from the figure, the peak stress decreased and then increased with the increase in polypropylene fiber addition, and the compressive strength increases and then decreases. The promoting effect of polypropylene fibers on peak stress did not become more significant with the increase in fiber content, although increasing the content of polypropylene fibers allowed more polypropylene fibers to be involved in crack resistance. However, doing so also increases the risk of clumping of polypropylene fibers, increasing the internal voids in the concrete and compromising the peak stress in the specimen. When the adverse effects of polypropylene fibers and their crack arresting effects cancel each other out, there is no significant change in peak stress.

The normalized peak stresses of all-coral seawater concrete with different fiber admixtures are given in Figure 18c. The data points in the figure are the ratios of peak stresses of all-coral seawater concrete specimens with different fiber contents of polypropylene fibers to those of all-coral seawater concrete specimens. As shown in the figure, the data points converge toward the value 1 when the fiber content is higher and the confining pressure value is higher. As a whole, the dispersion of the data points is more prominent at lower confining pressure values. The most significant dispersion was observed for the fiber content *V* = 2 kg·m^−3^. This shows that the different dosing of polypropylene fiber affects the peak stress. However, when the confining pressure value increased, the crack-arresting effect of polypropylene fiber was continuously weakened because the crack-arresting effect of the confining pressure value was more effective than that of polypropylene fiber. The fracture-stopping effect of the circumferential pressure value was more effective than that of polypropylene fiber, and the fracture-stopping effect of polypropylene fiber was weakened continuously as the circumferential pressure value increased. Polypropylene fibers were influenced by the combined positive effect of bridging and crack resistance and the negative effect of fiber agglomeration. To ensure the enhancement of the deformation performance of polypropylene fibers on all-coral seawater concrete to produce higher peak stresses at different values of circumferential pressure, it is recommended that the fiber dosing be equal to *V* = 3 kg·m^−3^.

### 4.2. Peak Strain

Figure 19 shows the relationship between the peak strain and the value of the confining pressure. As shown in the figure, the value of confining pressure was the main factor affecting the peak strain of the specimen. The peak strain of the specimen increased linearly with the increase in the value of the confining pressure. When *σ_w_* = 18 MPa, the peak strain of the specimen was 9.2 times that of the case without the constraint of the confining pressure value. On the one hand, the increase in peak strain was due to the confinement of the confining pressure value, which increased the elastic range of the specimen. The surface of coral aggregate is rough, and the specific surface area of porous structure is large. This made the cement matrix have a firm grip on the aggregate, and the bonding interface was strengthened significantly. On the other hand, the increased confining pressure value limited the development of microcracks to vertical penetration cracks. The plastic deformation of the specimens at peak strain also increased gradually due to the extrusion flow of cement mortar and the generation of fine cracks. The plastic deformation capacity of the polypropylene fiber all-coral seawater concrete increased substantially, which led to a significant increase in the peak strain of the specimens.

According to the experimental data, the formula for the peak strain of polypropylene fiber all-coral seawater concrete under different confining pressure values was fitted:(2)$$\frac{\varepsilon_{v}}{\varepsilon_{0}}=6.36\left(\frac{\sigma_{w}}{\sigma_{0}}\right)+0.81,\;R^{2}=0.96$$

*ε_0_* is the peak strain in the uniaxial compression state of the specimen.

The trend of peak strain for each group of specimens with different polypropylene fiber content is given in Figure 20. As can be seen from the figure, the peak strain gradually increased as the confining pressure value increased. When 0 MPa ≤ *σ_w_* ≤ 12 MPa, some specimens will be affected by local early damage to the weak zone when the oblique main crack mostly causes deformation of the specimen. In addition, the remaining portion of the specimen was less deformed, resulting in most of the polypropylene fibers bridging and the crack resistance not being fully developed. Therefore, the fiber content increased at this time, and the peak strain of the specimen grew more slowly. When 12 MPa ≤ *σ_w_* ≤ 18 MPa, the peak strain of the specimen increased significantly with the increase in the confining pressure value. Lateral restraint increased the friction between the complete coral aggregate and the cement matrix so that the deformation capacity of the specimen was further increased. Since the prominent cracks of the specimen gradually disappeared and the fine cracks increased, more polypropylene fibers inside the specimen were involved in crack-arresting so that the peak strain of the specimen increased.

Figure 21 shows the variation of polypropylene fiber dosage versus the mean value of peak strain of the specimens under each circumferential pressure. The peak strain increased and then decreased with the increase in polypropylene fiber addition, and the peak strain peaked at the fiber content *V* = 2 kg·m^−3^, at which time the peak strain increased by about 18% overall.

### 4.3. Elastic Modulus

Figure 22 shows the relationship between the elastic modulus of each group of specimens and the confining pressure value. As shown in the figure, the elastic modulus of the specimens grew nonlinearly as confining pressure value increased. The elastic modulus increase was more prominent when the confining pressure value was 0 MPa ≤ *σ_w_* ≤ 6 MPa. Next, the confining pressure value increased, and the rate of increase in elastic modulus gradually decreased because the confining pressure value constrained the lateral deformation of the specimen, resulting in an increasing elastic modulus of the specimen. However, as the confining pressure value increased, the internal voids of the concrete were continuously compressed at higher stress levels, resulting in a decrease in the rate of increase in the elastic modulus.

According to the experimental data, the formula for calculating the elastic modulus of polypropylene fiber all-coral seawater concrete under different confining pressure values was fitted:(3)$$\frac{E}{E_{0}}={\left(\frac{\sigma_{w}}{\sigma_{0}}\right)}^{0.66}+1,\;R^{2}=0.92$$

*E*_0_ is the elastic modulus of the specimen under uniaxial compression state.

Figure 23 shows the comparison of the effect of different polypropylene fiber addition on the elastic modulus of the specimens at the same confining pressure value. It can be seen that with the increase in the confining pressure value, the elastic modulus of the specimen increased nonlinearly. The optimum elastic modulus was obtained for polypropylene doping *V* = 3 kg·m^−3^ compared to other reinforced specimens.

As shown in Figure 24, the average modulus of elasticity decreased and then increased with the increase in fiber admixture. With the increase in confining pressure, the change range of average elastic modulus becomes smaller and smaller. Among them, the average elastic modulus of the specimens with fiber addition *V* = 3 kg·m^−3^ was the largest—14.9% higher than that with fiber content *V* = 1 kg·m^−3^. The average modulus of elasticity at fiber doping *V* = 2 kg·m^−3^ was 9% lower than that at *V* = 1 kg·m^−3^. With the addition of fibers, the elastic modulus was generally maintained at about 7 GPa after the peritectic pressure reached 18 MPa, an increase of 27% compared to the uniaxially compressed case. Its elastic modulus was higher than the concrete specimens without fiber admixture.

### 4.4. Ductility

In order to study the degree of influence of confining pressure value and fiber addition on the ductility performance of all-coral seawater concrete, the ductility coefficient *μ_c_* is used to express the ductility size of all-coral concrete, which is calculated as follows:(4)$$\mu_{c}=\frac{\varepsilon_{u}}{\varepsilon_{y}}$$

*ε_u_* denotes the strain corresponding to the end of damage or loading of the specimen; *ε_y_* denotes the yield strain.

A schematic diagram of the yield strength of the specimen using the isoenergetic method is shown in Figure 25. Table 7 shows the ductility coefficients of specimens with different fiber doping at different confining pressure values. It is observed in Figure 26 that the overall ductility coefficient of the specimens tended to increase and then decrease with the increase in the confining pressure value. The ductility coefficient of all-coral seawater concrete increased by 133% when *σ_w_* = 6 MPa compared to when *σ_w_* = 0 MPa. When *σ_w_* ≥ 6 MPa, the ductility coefficient started to decrease gradually but overall was still substantially higher than the uniaxial compression. The constraint effect of the confining pressure value improved the bearing capacity and deformation capacity of the specimen and increased the elastic interval of the specimen. The ductility coefficient measured at this stage was more accurate, and the faster growth of the ductility coefficient at this time indicates that the restraining effect of the confining pressure value can effectively improve the ductility of the all-coral concrete. When *σ_w_* ≥ 12 MPa, the deformation capacity of the specimen was further improved, and its stress–strain curve gradually leveled off after reaching the peak. The ductility coefficient shows a decreasing trend when the confining pressure value continues to increase, and the ductility coefficient stabilizes at about 1.45 when the confining pressure value is higher.

All specimens showed residual strengths that were more than 85% of the peak stress after loading, indicating that they are not damaged. Due to a rise in confining pressure value, the peak strain progressively approached the strain after the loading cycle, resulting in the computed ductility coefficient values’ convergence at *ε_v_*/0.7*ε_v_*. Therefore, the effect of its modification on the ductility coefficient of polypropylene-fiber-reinforced all-coral concrete after the confining pressure value reaches 12 MPa remains to be further investigated. In addition, the overall ductility coefficient of the specimens tended to decrease first and then increase with the increase in fiber content. From the overall analysis, the ductility improvement was the least for fiber admixture *V* = 2 kg·m^−3^, and the ductility of all-coral concrete specimens with fiber admixture *V* = 3 kg·m^−3^ was better.

### 4.5. Energy Consumption

The triaxial compression process of polypropylene-fiber-reinforced all-coral seawater concrete was accompanied by energy dissipation. The dissipated energy increased to a certain extent when the specimen released energy by generating and developing cracks in the form of thermal energy. In order to analyze the relationship between the actual energy consumption and the confining pressure and the amount of fiber, and to avoid the influence of the loading time on the accuracy of the energy consumption of the specimen, the integral area of the whole stress–strain curve is not used as the actual energy dissipation of the specimen. As shown in Figure 27, the integrated area of the stress–strain curve is taken as the actual energy dissipation (*Q*) when the specimen is damaged (the stress of the specimen is reduced to 85% of the peak stress). The actual energy consumption of the specimen is calculated using the following equation listed in Table 8, which is calculated as follows:(5)$$Q=\int_{0}^{\varepsilon_{u}}\sigma d\varepsilon$$

*ε_u_* denotes the longitudinal strain of the specimen when the bearing capacity is reduced to 0.85 times the peak stress, *σ* denotes the stress at each moment during the loading of the specimen, and *ε* denotes the strain of the specimen.

The function between the confining pressure value *σ_w_* and the actual energy consumption *Q* is obtained by fitting:(6)$$\frac{Q}{Q_{0}}=4.5\sigma_{w}^{0.8}+0.91$$

*Q*_0_ denotes the actual energy consumption of the specimen in uniaxial compression.

Figure 28a,b gives the relationship curves between the value of constraint pressure and fiber content and the actual energy consumption. From the figure, it can be seen that the relationship between the constraint pressure value and the actual energy consumption is nonlinear. As the confining pressure value increased, the actual energy consumption showed a trend of first stepping up and then leveling off. The actual energy consumption of all-coral seawater concrete specimens decreases and then increases as the fiber content increases. When *σ_w_* = 18 MPa, the energy consumption coefficient reached the maximum, which is about 39 times higher than that in the uniaxial compression state. The principal ways of consuming energy are the deformation of concrete components and the development of internal cracks. Thus, the specimen’s energy consumption increased mainly due to strength enhancement and the development of cracks. On one hand, the growth of energy consumption after mixing polypropylene fibers into the fiber skeleton restrained the lateral deformation of concrete, leading to an increase in the energy consumed by concrete due to elastic and plastic deformation. On the other hand, the fibers consumed energy in stretching and pulling to limit the development of cracks within the concrete.

Figure 29 shows the relationship between the energy consumption coefficient and the constraint pressure value and fiber dose. It reflects that the effect of constraint pressure value on energy consumption is significantly stronger than the effect of fiber dose on energy consumption. The energy consumption coefficient tended to stabilize when the confining pressure value increased to a certain level.

### 4.6. Damage Process Analysis

In order to describe the damage development process of polypropylene fiber all-coral aggregate seawater concrete specimens during loading, the damage development of the specimen was quantitatively analyzed. According to the continuum damage mechanics theory, the damage variable *D* is introduced to describe the degree of damage during the loading of the specimen, which is obtained according to the strain equivalence assumption [30].
(7)$$D=1-\frac{E^{*}}{E_{0}}$$

*σ*, *σ** are the effective stresses in the n-direction before and after the damage, respectively; *E*_0_ is the elastic modulus in the non-damaged state (taken as the secant modulus at the point where the stress in the rising section of the curve increases to 40% of the peak stress); *E** is the effective secant modulus after damage (taken as the secant modulus at each point on the stress–strain curve).

As shown in Figure 30a, it can be seen that the specimen was in a uniaxial compression state when *σ_w_* = 0 MPa. Before *ε* < 6 × 10^−3^, the axial load was small, the specimen was in the elastic stage with almost no damage, and the *D* value tended to zero. After *ε* > 6 × 10^−3^, the specimens showed initial damage, the damage coefficient increased linearly, and the damage occurred very rapidly. Once the load reached the peak stress, the cracks developed rapidly into vertical through-cracks. The axial bearing capacity of the specimen decreased rapidly, and damage occurred suddenly when the damage factor was close to 0.8. According to the analysis of Figure 31b, when the confining pressure value was 0 MPa ≤ *σ_w_* ≤ 6 MPa, with the increase in the confining pressure value, the strain of the specimen was more significant when the initial damage occurred, and the growth rate of the damage coefficient was obviously slowed down. This indicates that the presence of the constrained pressure value delayed the appearance of specimen damage. In addition, it delayed the appearance of specimen damage and effectively slowed down the development of specimen damage. When the confining pressure value was 6 MPa ≤ *σ_w_* ≤ 12 MPa, the specimen was already pressurized before the effective loading. This is because the target confining pressure value has a considerable significance value. This can cause some damage to the specimen before the effective loading. Comparing the curves in Figure 31b,c, it can be seen that the strain at the time of initial damage to the specimen decreased as the confining pressure value increased, and the development rate of the damage slowed down significantly after the specimen reached the peak strain. When the strain reached a specific value, the damage coefficient curves of the specimens with each fiber amount nearly overlapped, and the damage development trend was the same. When the confining pressure value 12 MPa ≤ *σ_w_* ≤ 18 MPa, the growth of the damage coefficient gradually tended to level off with the increasing strain.

The restraining effect of the confining pressure value effectively reduced the degree of damage to the specimen, slowed down the development process of damage to the specimen, and prevented the generation and development of cracks; even when the coral aggregate inside the specimen was brittle or crushed, evident damage phenomena did not occur within the specimen. Comparing Figure 30c,d with Figure 31c,d, can be seen that the initial damage strain of fiber-reinforced concrete was almost twice that of non-fiber-reinforced concrete.

## 5. Conclusions

(1)From the observation of the damaged specimens, the amount of polypropylene fiber addition and the confining pressure value have a significant influence on the mechanical strength of the specimens. The polypropylene fiber effectively reduced the crack width of the specimen and ensured the relative integrity of the specimen after damage. The crack pattern changed from vertical splitting damage to oblique splitting damage, and the width of the main crack decreased as the confining pressure value increased. The confining pressure value increased, and the crack pattern transformed into diagonal shear failure. When *σ_w_* ≥ 18 MPa, the crack pattern changed to transverse shear damage, the main crack of the specimen disappeared, and the residual deformation increased.(2)Confining pressure value improves the plastic deformation capacity of polypropylene fiber all-coral seawater concrete, reduces the risk of local concrete instability, prevents vertical penetration cracks, increases axial bearing capacity, and increases peak stress, peak strain, and elastic modulus. When the confining pressure reached 18 MPa, the peak value of the curve disappeared completely. The post-peak curve rose slowly, and the peak stress of the sample was 4.66 times that of uniaxial compression.(3)The microstructure of polypropylene fiber was observed by scanning electron microscope. It was found that the surface of polypropylene fiber was smooth, long, and thin. Coral aggregate has a large apparent density and internal friction angle, which is helpful in improving the interfacial adhesion between aggregate and hardened mortar. The strength of concrete has been improved to a certain extent. Fiber is like microreinforcement, and its random and uniform three-dimensional spatial distribution can prevent the aggregate from sinking in the concrete. The fiber may bridge fractures and enhance the interface transition zone’s stability. It partially compensates for concrete imperfections and regulates the plastic shrinkage of concrete, effectively preventing the initiation and propagation of microcracks in fresh concrete.(4)When *σ_w_* = 6 MPa, the ductility coefficient of all-coral seawater concrete increases by 133% compared with that of *σ_w_* = 0 MPa; when *σ_w_* ≥ 6 MPa, the ductility coefficient begins to decrease gradually, but the overall is still substantially higher than the uniaxial compression. The ductility increase is the smallest when the fiber content *V* = 2 kg·m^−3^.(5)The actual energy consumption increases and then levels off as the confining pressure value increases. When 0 MPa ≤ *σ_w_* ≤ 6 MPa, and with the increase in fiber addition, the actual energy consumption shows a trend of first increasing and then decreasing, and with the further increase in the confining pressure value, the actual energy consumption tends to be stable. When *σ_w_* = 18 MPa, the energy consumption coefficient reaches the maximum, about 39 times higher than that in the uniaxial compression state.(6)In general, the specimen’s damage coefficient curve becomes more oblong when the confining pressure value is raised. The later the initial damage appears, the slower the damage development process and the lower the damage degree. When 12 MPa ≤ *σ_w_* ≤ 18 MPa, the strain of fiber-doped concrete when the initial damage occurs is almost twice the initial damage strain of fiberless concrete.

## Figures and Tables

**Figure 1 materials-15-04234-f001:**
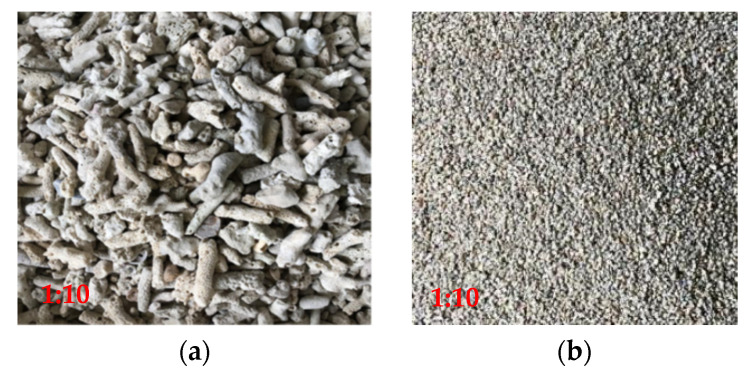
Coral aggregate. (**a**) Coral coarse aggregates. (**b**) Coral fine aggregates.

**Figure 2 materials-15-04234-f002:**
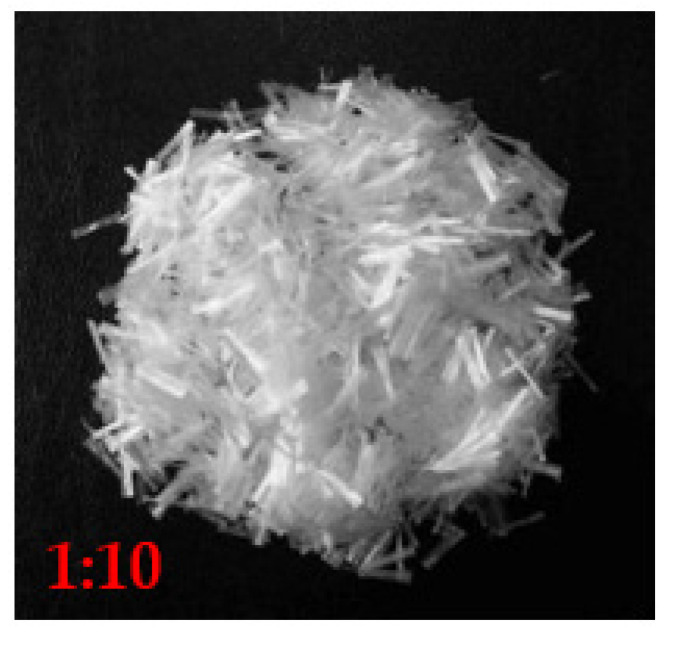
Polypropylene fiber.

**Figure 3 materials-15-04234-f003:**
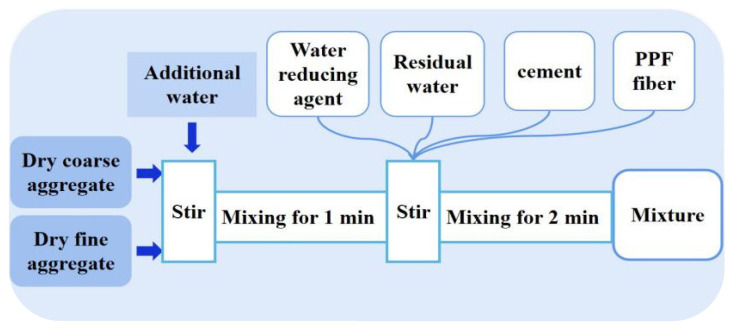
Dosing sequence diagram.

**Figure 4 materials-15-04234-f004:**
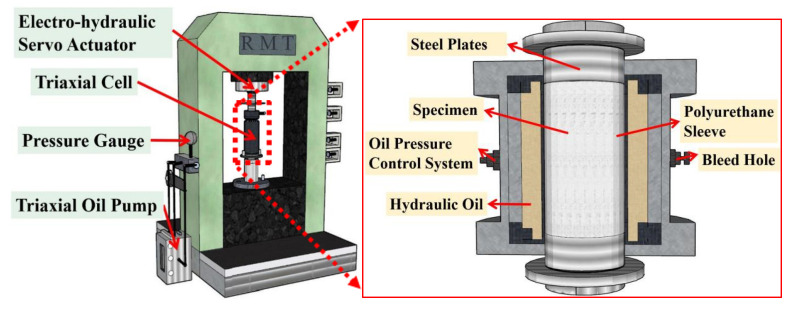
Loading device.

**Figure 5 materials-15-04234-f005:**
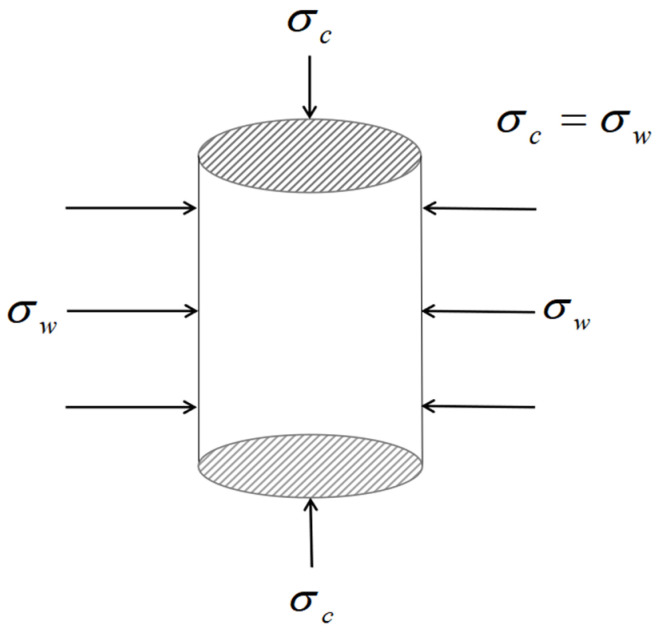
Test force model.

**Figure 6 materials-15-04234-f006:**
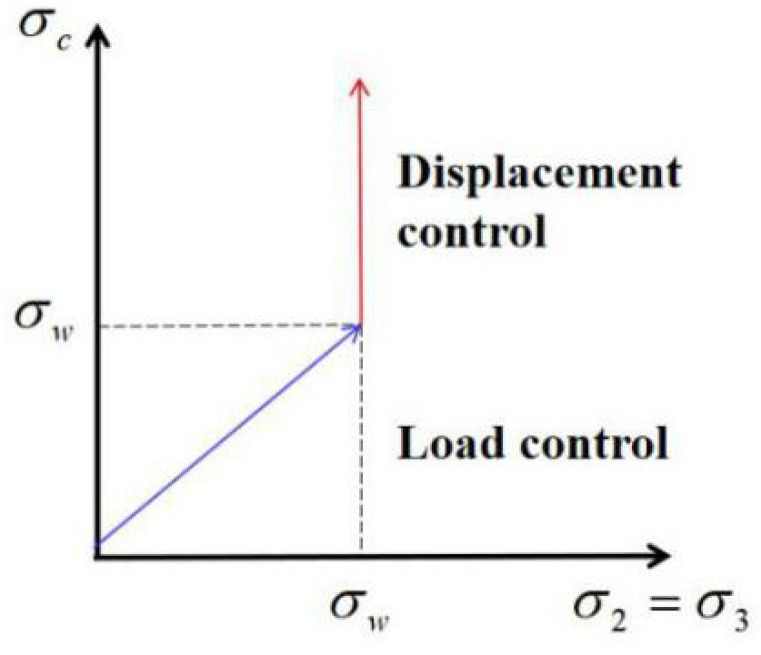
Loading system.

**Figure 7 materials-15-04234-f007:**
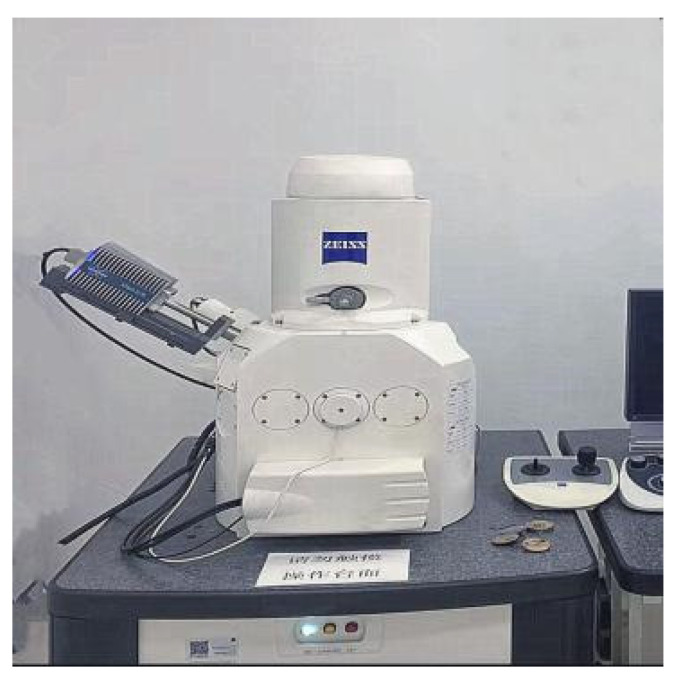
GeminiSEM 360.

**Figure 8 materials-15-04234-f008:**
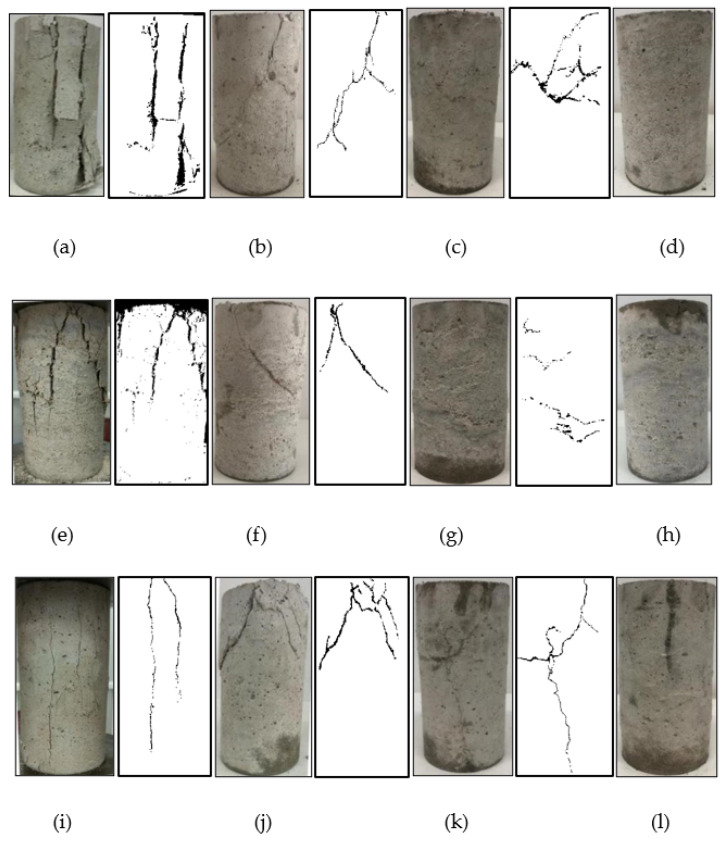
Destructive form of specimen. (**a**) PF-1-0-A, (**b**) PF-1-6-A, (**c**) PF-1-12-A, (**d**) PF-1-18-A, (**e**) PF-2-0-A, (**f**) PF-2-6-A, (**g**) PF-2-12-A, (**h**) PF-2-18-A, (**i**) PF-3-0-A, (**j**) PF-3-6-A, (**k**) PF-3-12-A, (**l**) PF-3-18-A.

**Figure 9 materials-15-04234-f009:**
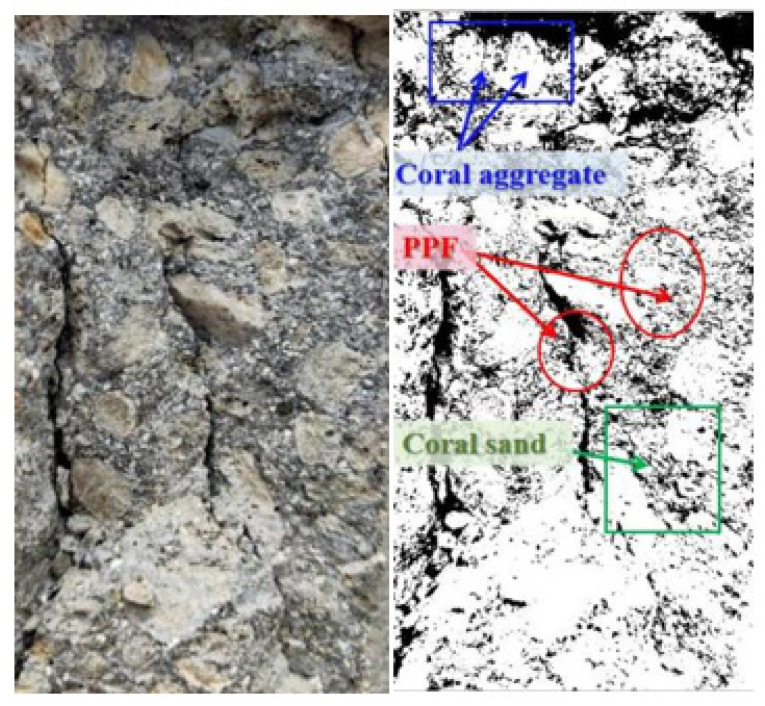
Distribution of fibers and aggregates in longitudinal section.

**Figure 10 materials-15-04234-f010:**
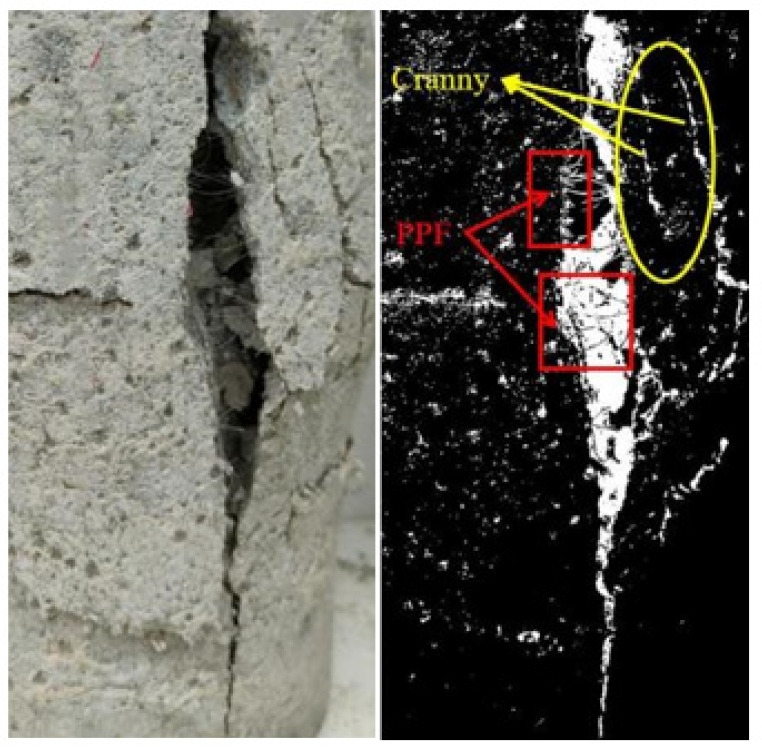
Bridging effect of PPF during triaxial failure of specimens.

**Figure 11 materials-15-04234-f011:**
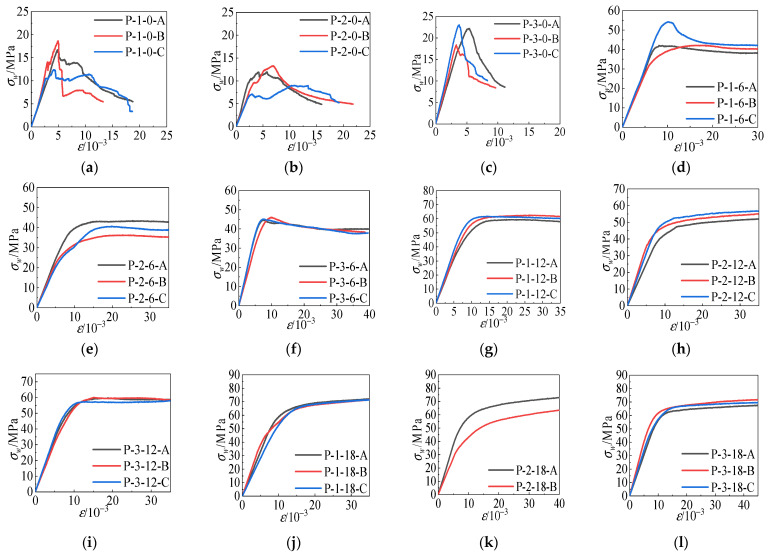
Full stress–strain curve of specimen. (**a**) PF-1-0, (**b**) PF-2-0, (**c**) PF-3-0, (**d**) PF-1-6, (**e**) PF-2-6, (**f**) PF-3-6, (**g**) PF-1-12, (**h**) PF-2-12, (**i**) PF-3-12, (**j**) PF-1-18, (**k**) PF-2-18, (**l**) PF-3-18.

**Figure 12 materials-15-04234-f012:**
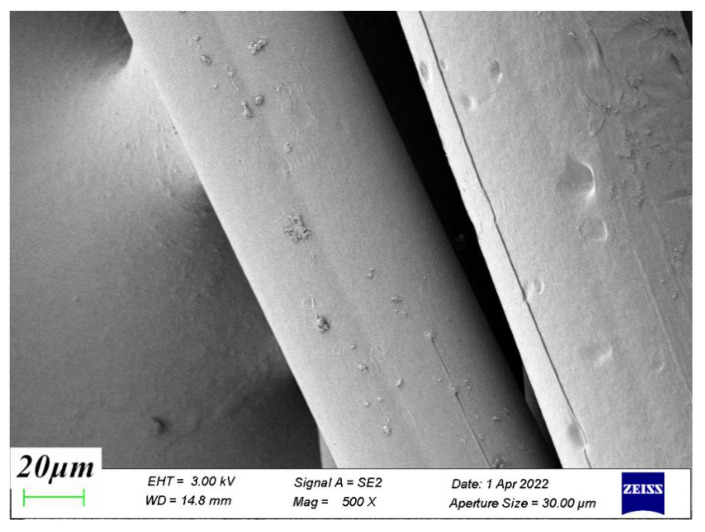
Polypropylene fiber.

**Figure 13 materials-15-04234-f013:**
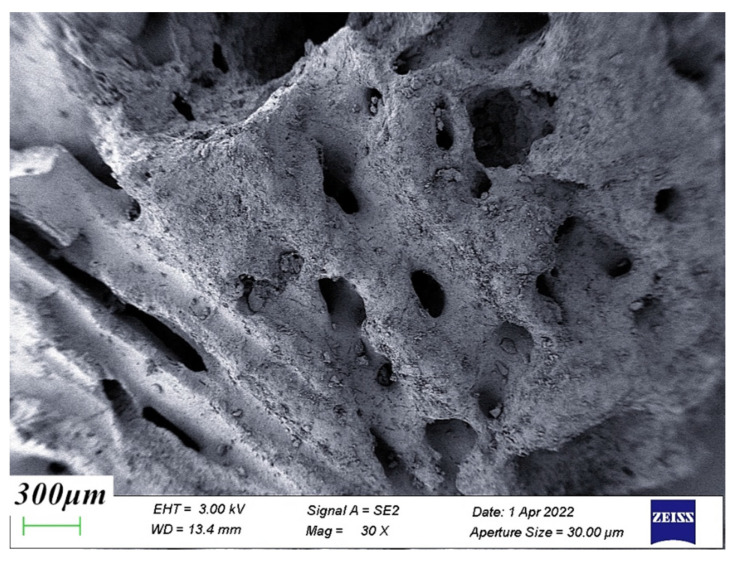
Coral coarse aggregate.

**Figure 14 materials-15-04234-f014:**
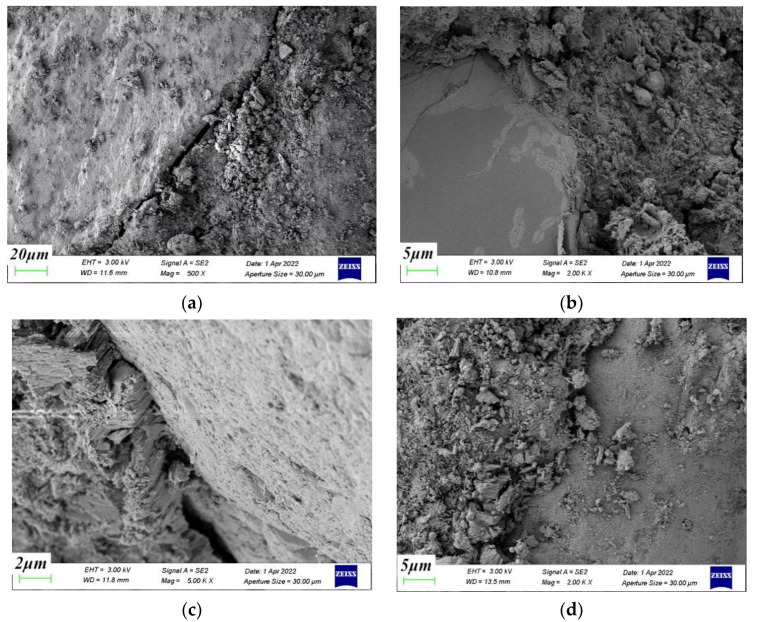
Coral aggregate–slurry interface transition zone. (**a**) 0 MPa, (**b**) 6 MPa, (**c**) 12 MPa, (**d**) 18 MPa.

**Figure 15 materials-15-04234-f015:**
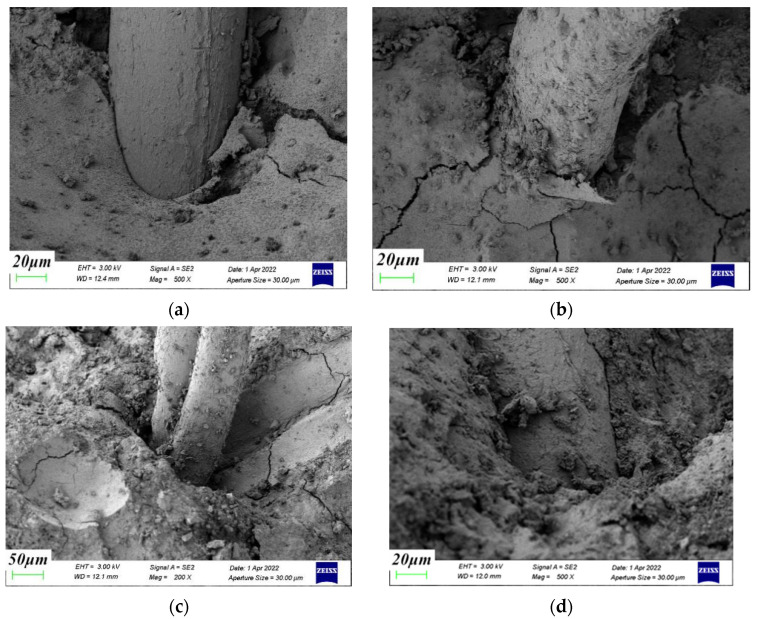
Fiber–cement paste interface transition zone. (**a**) 0 MPa, (**b**) 6 MPa, (**c**) 12 MPa, (**d**) 18 MPa.

**Figure 16 materials-15-04234-f016:**
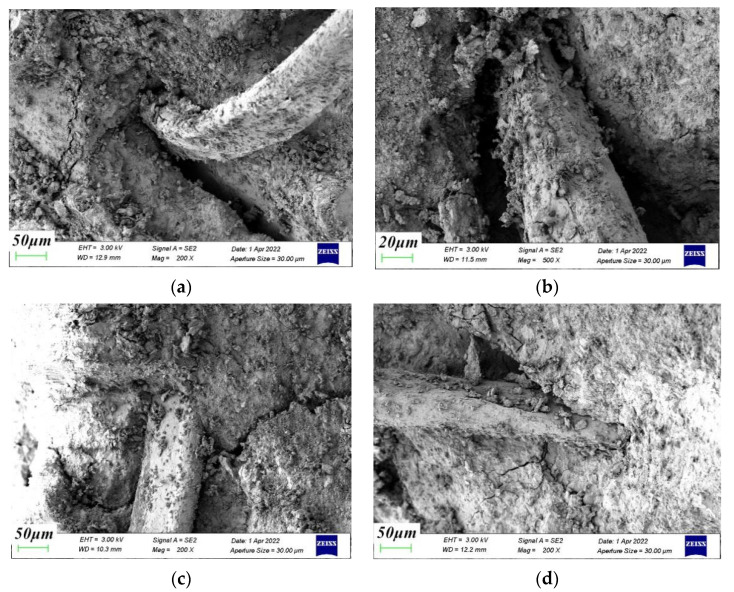
Transition zone of fiber–aggregate interface. (**a**) 0 MPa, (**b**) 6 MPa, (**c**) 12 MPa, (**d**) 18 MPa.

**Figure 17 materials-15-04234-f017:**
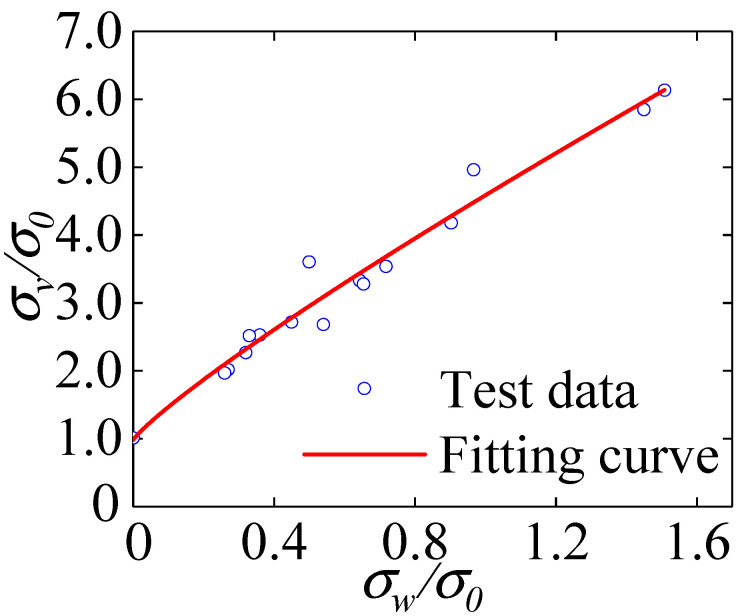
Relationship between peak stress and confining pressure value.

**Figure 18 materials-15-04234-f018:**
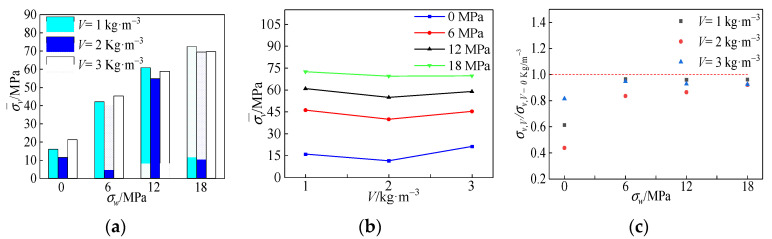
Effect of polypropylene fiber on peak stress: (**a**) influence of polypropylene fiber on peak stress; (**b**) relationship between peak stress and polypropylene fiber content; (**c**) normalized peak stress.

**Figure 19 materials-15-04234-f019:**
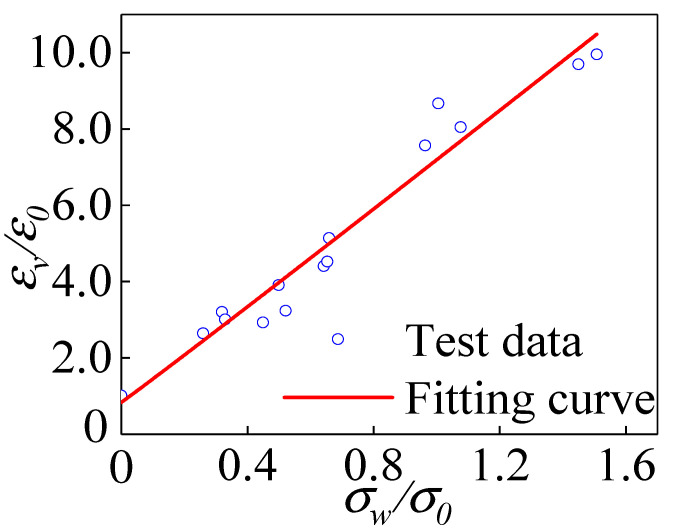
Relationship between peak strain and confining pressure value.

**Figure 20 materials-15-04234-f020:**
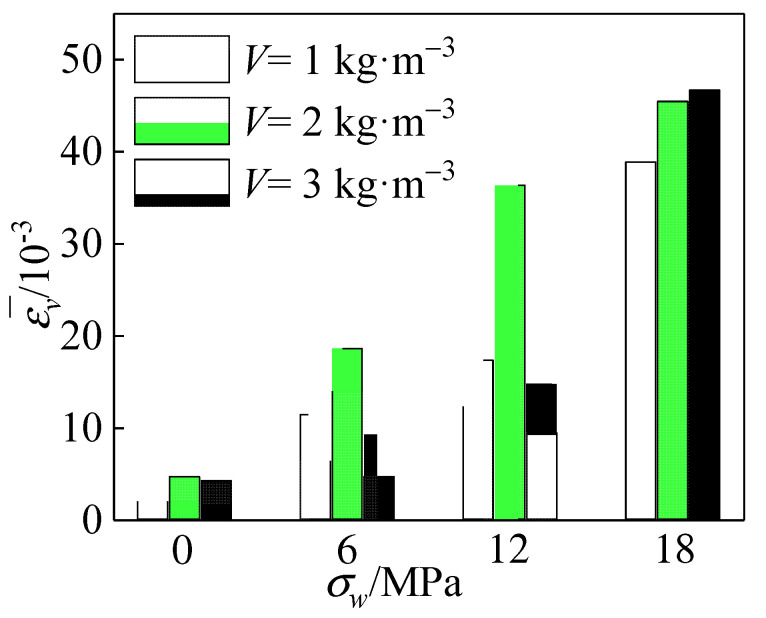
Effect of polypropylene fibers addition on peak strain.

**Figure 21 materials-15-04234-f021:**
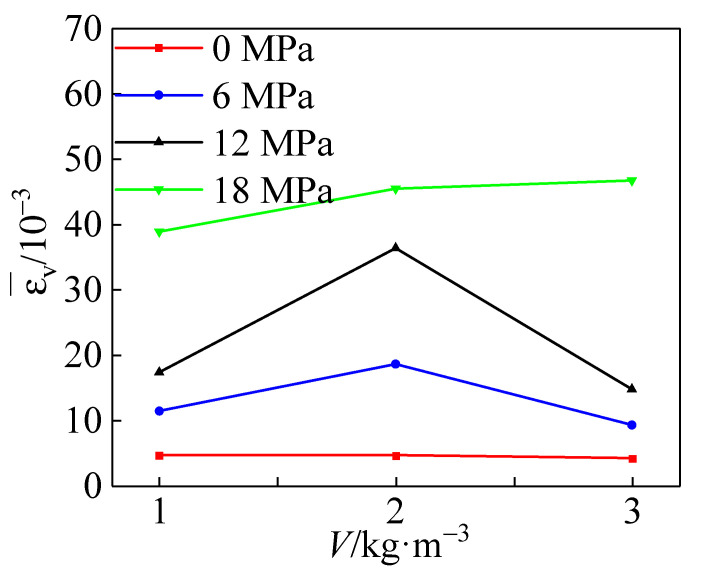
Relationship between the amount of polypropylene fibers addition and the average peak strain.

**Figure 22 materials-15-04234-f022:**
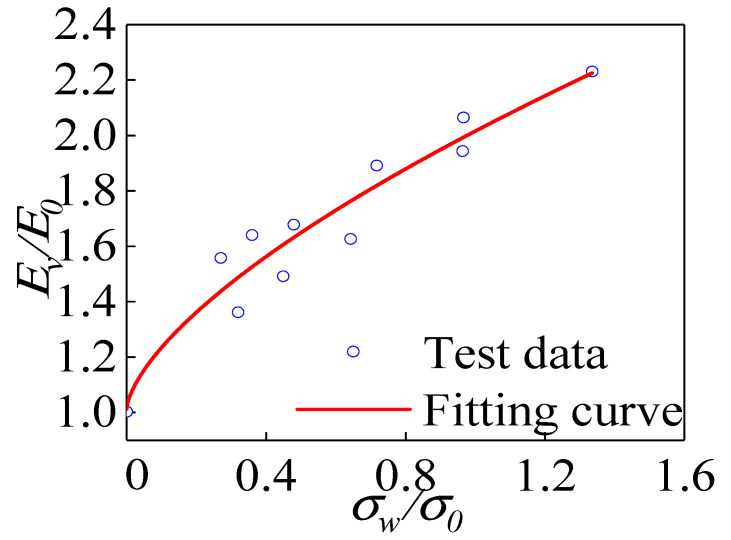
Relationship between elastic modulus and confining pressure value.

**Figure 23 materials-15-04234-f023:**
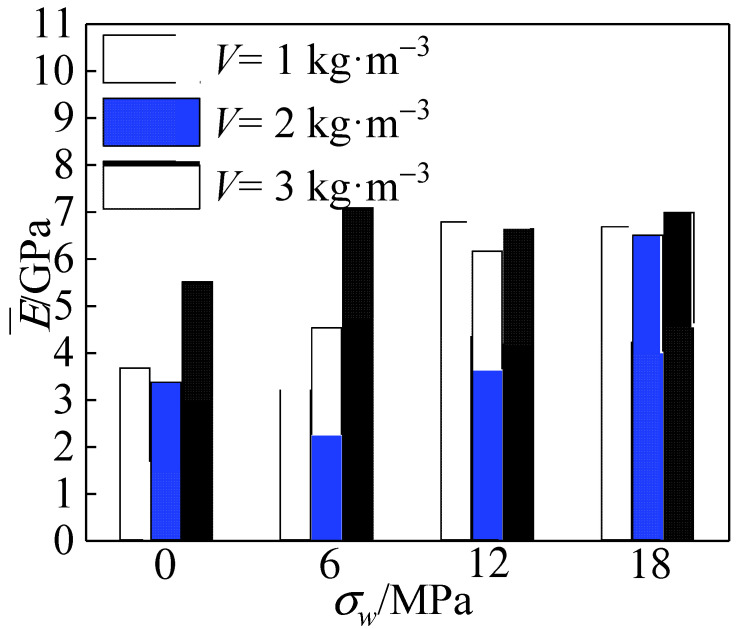
Effect of polypropylene fiber admixture on elastic modulus.

**Figure 24 materials-15-04234-f024:**
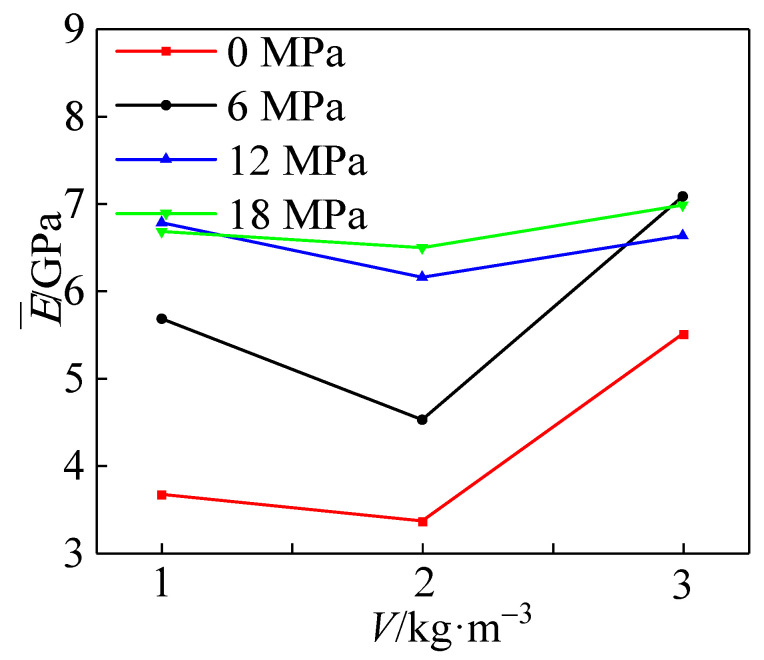
Effect of polypropylene fiber admixture on the average elastic modulus.

**Figure 25 materials-15-04234-f025:**
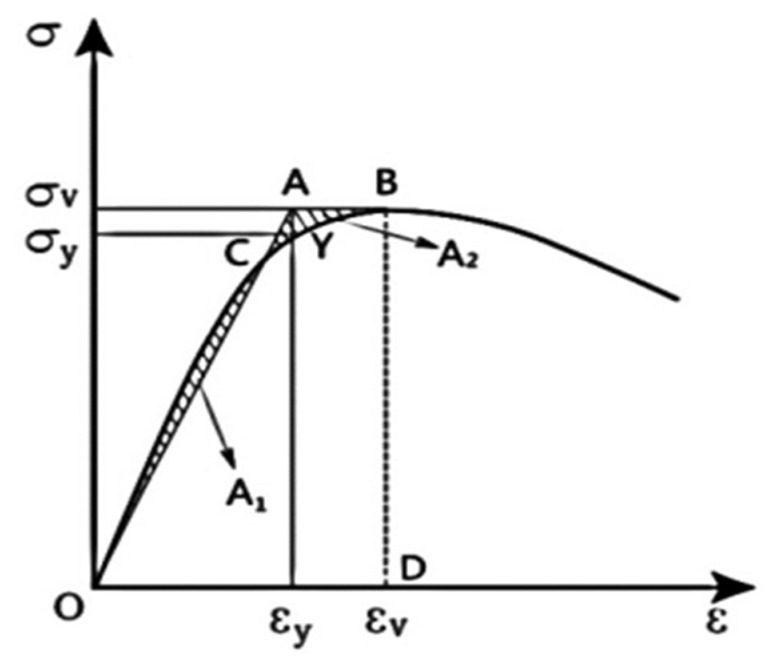
Isoenergetic method to find the yield strength.

**Figure 26 materials-15-04234-f026:**
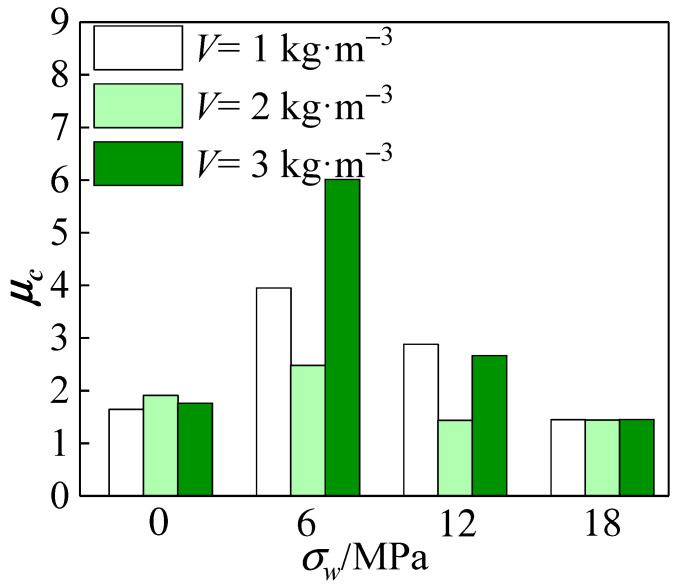
Relationship between ductility coefficient and confining pressure value.

**Figure 27 materials-15-04234-f027:**
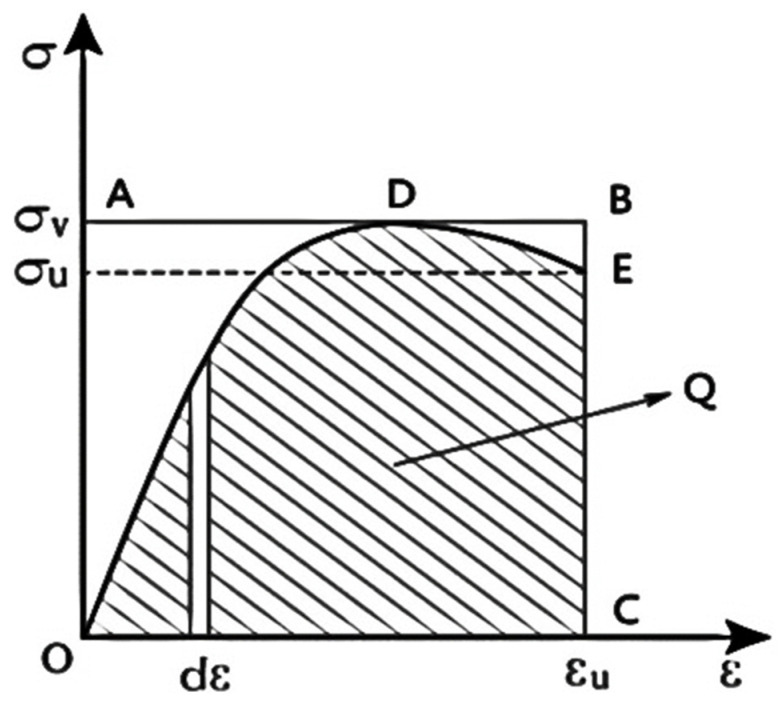
Schematic diagram of energy consumption calculation [29].

**Figure 28 materials-15-04234-f028:**
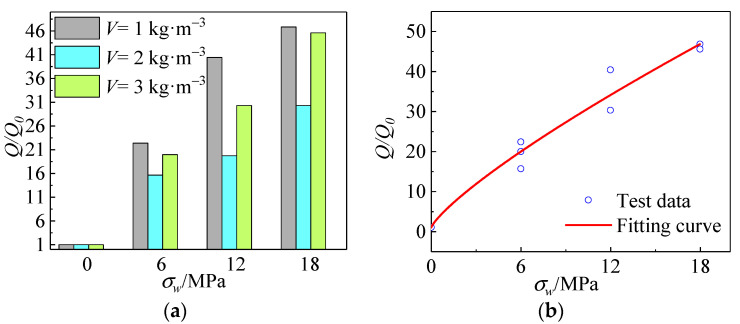
(**a**) Relationship between actual energy consumption and confining pressure value and fiber content; (**b**) fitting curve between actual energy consumption and confining pressure value.

**Figure 29 materials-15-04234-f029:**
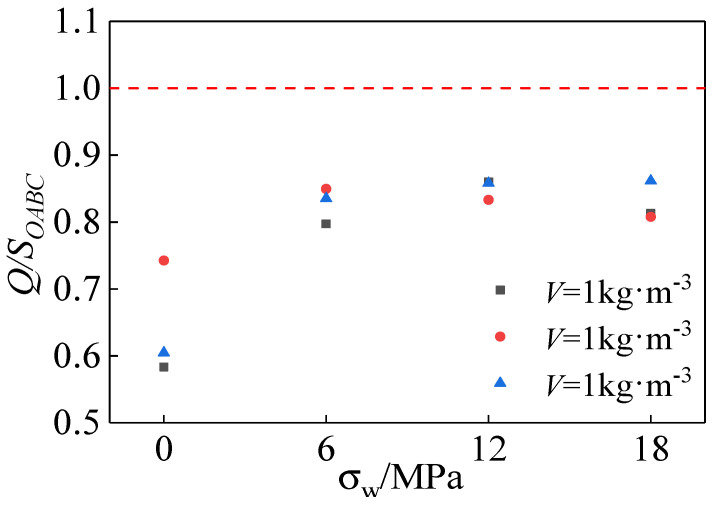
Relationship between energy consumption coefficient and confining pressure value and fiber content.

**Figure 30 materials-15-04234-f030:**
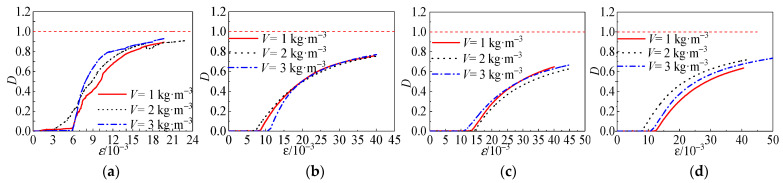
Effect of different variation parameters on damage: (**a**) *σ_w_* = 0 MPa, (**b**) *σ_w_* = 6 MPa, (**c**) *σ_w_* = 12 MPa, and (**d**) *σ_w_* = 18 MPa.

**Figure 31 materials-15-04234-f031:**
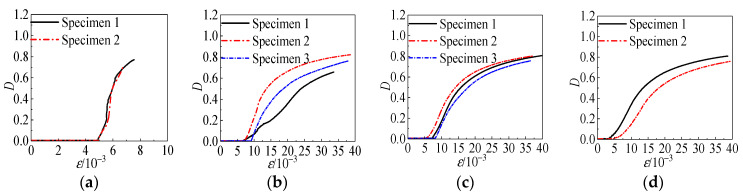
Damage factor of fiberless all-coral seawater concrete versus different variations. Parameters: (**a**) *σ_w_* = 0 MPa, (**b**) *σ_w_* = 6 MPa, (**c**) *σ_w_* = 12 MPa, and (**d**) *σ_w_* = 18 MPa.

**Table 1 materials-15-04234-t001:** Coral aggregate basic physical properties index.

Classification	Apparent Density (kg·m^−3^)	Stacking Density (kg·m^−3^)	Dimension/mm	Porosity (%)	Shell Content (%)	1 h Water Absorption Rate (%)	Cl^−^ Content (%)	Cylinder Compression Strength (MPa)	Fineness Modulus
Coral Coarse Aggregate	1957.50	893.75	40–60	54.21	25.58	8.75%	0.02	4.00	
Coral sand	1197.90	1055.00	2.5–5	41.51	31.72	19.40	0.21		2.85

**Table 2 materials-15-04234-t002:** Basic parameters of polypropylene fibers.

Type	Length (mm)	Diameter (μm)	Tensile Strength (MPa)	Elastic Modulus (MPa)	Elongation at Break (%)
Polypropylene fiber	18.0	32.7	469.0	4236.0	28.4

**Table 3 materials-15-04234-t003:** Chemical composition of natural seawater.

Compound	NaCl	MgCl_2_	Na_2_SO_4_	CaCl_2_	KCl	NaHCO_3_	KBr
Content(%)	1.070	0.220	0.220	0.053	0.053	0.007	0.007

**Table 4 materials-15-04234-t004:** Concrete mix ratio design.

Strength Grade	Coarse Aggregate (kg·m^−3^)	Sand (kg·m^−3^)	Cement (kg·m^−3^)	Water (kg·m^−3^)	Additional Water (kg·m^−3^)	Water Reducer (%)	Net Water–Ash Ratio	Total Water–Ash Ratio
C30	749.00	749.00	613.00	215.20	119.80	0.20	0.35	0.55

**Table 5 materials-15-04234-t005:** Specimen design parameters.

Specimen Number	Strength Grade	*σ_w_* (MPa)	*V* (kg/m^3^)	Specimen Number	Strength Grade	*σ_w_* (MPa)	*V* (kg/m^3^)
PF-1-0-A	C30	0	1	PF-1-12-A	C30	12	1
PF-1-0-B	C30	0	1	PF-1-12-B	C30	12	1
PF-1-0-C	C30	0	1	PF-1-12-C	C30	12	1
PF-2-0-A	C30	0	2	PF-2-12-A	C30	12	2
PF-2-0-B	C30	0	2	PF-2-12-B	C30	12	2
PF-2-0-C	C30	0	2	PF-2-12-C	C30	12	2
PF-3-0-A	C30	0	3	PF-3-12-A	C30	12	3
PF-3-0-B	C30	0	3	PF-3-12-B	C30	12	3
PF-3-0-C	C30	0	3	PF-3-12-C	C30	12	3
PF-1-6-A	C30	6	1	PF-1-18-A	C30	18	1
PF-1-6-B	C30	6	1	PF-1-18-B	C30	18	1
PF-1-6-C	C30	6	1	PF-1-18-C	C30	18	1
PF-2-6-A	C30	6	2	PF-2-18-A	C30	18	2
PF-2-6-B	C30	6	2	PF-2-18-B	C30	18	2
PF-2-6-C	C30	6	2	PF-2-18-C	C30	18	2
PF-3-6-A	C30	6	3	PF-3-18-A	C30	18	3
PF-3-6-B	C30	6	3	PF-3-18-B	C30	18	3
PF-3-6-C	C30	6	3	PF-3-18-C	C30	18	3

**Table 6 materials-15-04234-t006:** Feature point parameters.

Specimen No.	Yield Point		Peak Point		Destruction Point		Elastic Modulus *E* (GPa)
	*P_y_* (kN)	Δ*_y_* (mm)	*P_m_* (kN)	Δ*_m_* (mm)	*P_u_* (kN)	Δ*_u_* (mm)	
PF-1-0-A	16.27	4.88	16.70	4.84	14.20	6.32	3.38
PF-1-0-B	17.66	4.62	18.65	4.96	15.85	5.06	4.03
PF-1-0-C	11.13	3.51	12.40	4.14	10.54	4.68	3.59
PF-2-0-A	10.71	3.21	11.956	5.75	10.16	8.29	4.23
PF-2-0-B	12.01	5.03	13.28	6.87	11.29	7.98	3.04
PF-2-0-C	6.33	6.41	9.00	10.85	7.65	15.31	2.82
PF-3-0-A	21.09	4.71	22.20	5.40	18.87	6.26	4.89
PF-3-0-B	18.32	3.27	18.33	3.34	15.58	4.77	5.31
PF-3-0-C	22.47	3.54	22.99	3.80	19.54	4.24	6.32
PF-1-6-A	40.86	7.38	42.04	8.12	39.29	38.52	5.54
PF-1-6-B	37.72	8.82	42.17	18.43	40.67	39.22	5.48
PF-1-6-C	52.35	8.87	54.22	10.17	46.09	14.47	6.02
PF-2-6-A	39.67	10.14	43.46	25.45	42.23	39.30	4.91
PF-2-6-B	31.86	10.35	36.32	22.82	35.26	39.26	4.53
PF-2-6-C	35.55	12.58	40.69	19.89	38.93	38.74	4.13
PF-3-6-A	41.92	6.11	44.73	7.54	39.90	39.77	7.61
PF-3-6-B	42.54	7.76	46.02	10.00	39.11	31.81	6.15
PF-3-6-C	42.64	6.12	45.08	7.65	38.32	31.89	7.49
PF-1-12-A	53.62	10.86	59.45	19.73	57.55	38.00	6.39
PF-1-12-B	57.39	10.76	62.57	26.33	61.11	38.29	6.55
PF-1-12-C	56.88	8.90	61.78	14.38	60.07	38.20	7.41
PF-2-12-A	47.33	13.33	52.16	35.80	51.97	36.22	4.99
PF-2-12-B	49.08	11.73	55.47	37.88	55.39	38.06	7.19
PF-2-12-C	51.49	11.65	56.90	35.86	56.66	35.87	6.29
PF-3-12-A	54.16	10.04	59.71	18.80	58.49	41.30	6.72
PF-3-12-B	54.74	10.73	60.00	15.08	58.02	36.25	6.12
PF-3-12-C	56.43	10.79	59.10	38.63	59.10	38.63	7.06
PF-1-18-A	65.47	14.15	72.87	38.54	72.54	39.08	6.63
PF-1-18-B	63.81	14.27	72.33	38.06	72.07	38.35	7.83
PF-1-18-C	66.00	16.01	72.51	40.15	71.77	41.13	5.58
PF-2-18-A	64.00	14.80	73.24	41.70	73.13	41.75	7.49
PF-2-18-B	56.34	20.96	65.88	49.59	65.66	50.63	5.50
PF-2-18-C							
PF-3-18-A	62.92	14.44	68.04	50.58	67.92	51.10	6.06
PF-3-18-B	64.82	12.99	72.00	47.69	71.94	49.42	8.15
PF-3-18-C	64.27	13.62	70.038	47.64	69.71	47.78	6.74

Note: *P_y_* and Δ*_y_* denote the horizontal force and horizontal displacement at the nominal yield point, respectively, determined by the energy equivalence method; *P_m_* and Δ*_m_* denote the horizontal force and displacement at the peak point, respectively; *P_u_* and Δ*_m_* horizontal force and displacement at the damage point; the damage point is defined as the point corresponding to 85% of the peak load in the descending section of the skeleton; *E* denotes the elastic modulus.

**Table 7 materials-15-04234-t007:** Ductility factor.

μ	P-0	P-6	P-12	P-18
P-1 kg·m^−3^	1.65	3.95	2.88	1.45
P-2 kg·m^−3^	1.91	2.48	1.44	1.44
P-3 kg·m^−3^	1.76	6.01	2.67	1.45

**Table 8 materials-15-04234-t008:** Actual energy consumption.

Q	P-0 MPa	P-6 MPa	P-12 MPa	P-18 MPa
P-1 kg·m^−3^	49.75	1114.33	2010.98	2330.77
P-2 kg·m^−3^	85.11	1333.96	1678.90	2580.25
P-3 kg·m^−3^	65.38	1305.12	1981.80	2981.45

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
