# Peer review of "Experimental Study on Triaxial Compressive Mechanical Properties of Polypropylene Fiber Coral Seawater Concrete"

_materials, 2022, doi:10.3390/ma15124234_

Round 1

Reviewer 1 Report

comments in pdf file

Author Response

Response to Reviewer 1 Comments

Dear reviewers:

First of all, thank you very much for taking the time to review our manuscript entitled “Experimental study on triaxial compressive mechanical properties of polypropylene fiber coral seawater concrete (materials-1728110)”. These comments are all valuable and very helpful for revising and improving our paper, as well as having important guiding significance to our researches.

We have studied comments carefully and have made correction which we hope meet with approval. Revised portions are marked in red in the paper. The related statements for these revisions are listed as follows.

Lastly, thanks very much for your fruitful work and patience.

Yours sincerely,

Hang Shi, Linlin Mo, Mingyan Pan, Leiguo Liu, Zongping Chen *.

May. 26, 2022

Point 1: Text position: 98" (GB/T 17431.2-2010)  

add all tests to references  

Response 1: Thanks for your suggestions. We have carefully studied the references used in this paper and have added relevant statements and discussions in some corresponding places. The modified contents are marked in red font in the text.

The supplementary literature is as follows:

[24]China Building Materials Federation (2010) GB/T 17431.2-2010. General Administration of quality supervision, inspection and Quarantine of the people's Republic of China; China National Standardization Administration Committee.

[25]Ministry of construction of the people's Republic of China Industrial standard of the people's Republic of China JGJ 12-2006 / Technical Specification for lightweight aggregate concrete structures [M] China Construction Industry Press, 2006.

[26]GB500010-2010,Code for design concrete structures[S]. Beijing: China Architecture & Building Press, 2010. (in Chinese).

Point 2: Text position: (a) Coral coarse aggregates

add scale to fotos, it is important to see dimension of elements, you can add on area of foto enlargement elements.  

Response 2: Thanks for your suggestions and questions. We have added relevant contents and marked in red font in the corresponding part of the text.

Point 3: Text position: Table 1. Coral aggregate basic physical properties index

add dimensions of aggregates.

Response 3: Thanks for your suggestions and questions. We have added relevant contents and marked in red font in the corresponding part of the text.

Point 4: Text position: Table 2. Basic parameters of polypropylene fibers

source of data and mathods of analysis

Response 4: Thanks for your suggestions and questions. The experimental method and data are obtained from the inspection of Beijing Zhongke Institute of photoanalytical Chemical Technology (chemical laboratory).

Avio200 Inductively coupled plasma emission spectrometer ((ICP-OES)Avio200

Dx-300st Solid liquid powder three purpose densimeter

Cic-d100 Ion chromatograph

Phs-3c PH meter

Point 5: Text position: Table 3. Chemical composition of natural seawater

source of data and methods of analysis

Response 5: Thanks for your suggestions and questions.The experimental method and data are obtained from the inspection of Beijing Zhongke Institute of photoanalytical Chemical Technology (chemical laboratory).

Avio200 Inductively coupled plasma emission spectrometer ((ICP-OES)Avio200

Dx-300st Solid liquid powder three purpose densimeter

Cic-d100 Ion chromatograph

Phs-3c PH meter

Point 6: Text position: Table 4. Concrete mix ratio design

change no. of tab, in the text too

Response 6: Thanks for your suggestions and questions. Table 4 is equivalent to the mix proportion of foundation full coral aggregate seawater concrete. We designed a total of 36 test blocks. Among them, 0mpa, 6Mpa, 12MPa and 18Mpa are taken as confining pressure values. The fiber contents were (1kg·m-3, 2kg·m-3, 3kg·m-3) respectively. For example, in PF-1-0-A, PF represents fiber, 1 represents fiber content 1kg·m-3, 0 represents 0MPa, A, B and C represent standard group, and there are three identical test blocks in a group. The confining ratio of the test block is different from that of the coral concrete, and only the confining ratio of the test block is the same.

Point 7: Text position: Table 5. Specimen design parameters

change no. of tab, in the text too

Response 7: Thanks for your suggestions and questions.According to table 5, we rechecked figure 7 destructive form of specimen and rearranged the number of test blocks. Check and unify the test block codes in figure 8, figure 11 and table 6.

Point 8: Text position: 3 Experimental results and analysis

add software reference

Response 8: Thanks for your suggestions and questions.We have added relevant contents and marked in red font in the corresponding part of the text.

The supplementary software literature is as follows:

Liu C. *, Tang C., Shi B., Suo W., 2013. Automatic quantification of crack patterns by image processing. Computers and Geosciences, 57, 77-80. [doi: 10.1016/j.cageo.2013.04.008]

Point 9: Text position: (a) PF-A-1-0

      nos of speciments in tab 5 are different, is it another speciments

Response 9: Thanks for your suggestions and questions.Yes, we have designed a total of 36 test blocks. Among them, 0mpa, 6Mpa, 12MPa and 18Mpa are taken as confining pressure values. The fiber contents were (1kg·m-3, 2kg·m-3, 3kg·m-3) respectively. For example, in PF-1-0-A, PF represents fiber, 1 represents fiber content 1kg·m-3, 0 represents 0MPa, A, B and C represent standard group, and there are three identical test blocks in a group. The confining ratio of the test block is different from that of the coral concrete, and only the confining ratio of the test block is the same.

Point 10: Text position: Figure 12. Polypropylene fiber

Fig. 12 is existing, change nos of figs in the text, see value of fiber diameter in tab. 2 and on te image, is it correct?

Response 10: Thanks for your suggestions and questions. We are very sorry for this type of

error. Although different instruments are used to detect its diameter, it may be the problem of the magnification and reduction mode of SEM observation. Perhaps this SEM polypropylene fiber picture observed at the same time can better express the size of fiber diameter. It conforms to the polypropelene fiber diameter (32.7) measured by Beijing Zhongke Institute of light analysis and Chemical Technology (chemical laboratory) , we replaced the picture.Thank the editor for finding the problem carefully.

Point 11: Text position: Figure 13. Coral coarse aggregate

scanning electron microscopy (SEM), add to method discription

Response 11: Thanks for your suggestions and questions. We have added relevant contents

and marked in red font in the corresponding part of the text.

Point 12: Text position: 355 add a, b, c to part of figure

Response 12: Thanks for your suggestions. We have added relevant contents and marked in red font in the corresponding part of the text.

Point 13: Text position: Figure.27 Schematic diagram of energy consumption calculation

reference fig. 27 in the text

Response 13: Thanks for your suggestions. We have added relevant contents and marked in red font in the corresponding part of the text. Supplementary references are as follows:

Jidong Cui,“HLA:Hysteretic Loop Analysis Program", http:/www.idcui.com/?p=11828.

Point 14: Text position: Figure 28. The relationship between the actual energy consumption and the confining pressure value and fiber doping.

describe a and b in caption

Response 14: Thanks for your suggestions. We have added relevant explanations in the text and marked the corresponding parts in the text in red font.

Point 15: Text position: Figure 30. Effect of different variation parameters on damage:

(a)σw=0MPa(b)σw=6MPa(c)σw=12MPa(d)σw=18MPa

Response 15: Thanks for your suggestions. We have added relevant explanations in the text and marked the corresponding parts in the text in red font.

Point 16: Text position: Figure 31. Damage factor of fiber less all-coral seawater concrete versus different 

variation parameters: (a)σw=0MPa(b)σw=6MPa(c)σw=12MPa(d)σw=18MPa

Response 16: Thanks for your suggestions. We have added relevant explanations in the text and marked the corresponding parts in the text in red font.

Reviewer 2 Report

The authors have presented some interesting data on the mechanical properties of coral seawater concrete with polypropylene fiber. Following are a few comments for this paper;

1- The introduction should include references from the latest papers

2- What was the reason behind the selection of the fiber with specific dimensions/ aspect ratio. Need some clarity

3- Please include references for the standard tests

4- There are no papers cited in the discussion section. Please include some papers  to improve your discussion and also compare them with some other studies

5- Overall, the paper has some interesting data, but needs major improvement in written English. There are many sentences forming errors that should be corrected before publication

Author Response

Response to Reviewer 2 Comments

Dear reviewers:

First of all, thank you very much for taking the time to review our manuscript entitled “Experimental study on triaxial compressive mechanical properties of polypropylene fiber coral seawater concrete (materials-1728110)”. These comments are all valuable and very helpful for revising and improving our paper, as well as having important guiding significance to our researches.

We have studied comments carefully and have made correction which we hope meet with approval. Revised portions are marked in red in the paper. The related statements for these revisions are listed as follows.

Lastly, thanks very much for your fruitful work and patience.

Yours sincerely,

Hang Shi, Linlin Mo, Mingyan Pan, Leiguo Liu, Zongping Chen *.

May. 26, 2022

Point 1: The introduction should include references from the latest papers.

Response 1: Thanks for your suggestions. We have carefully studied the references used in this paper and have added relevant statements and discussions in some corresponding places. The modified contents are marked in red font in the text.

We added three articles that the fourth reviewer hoped to add to enrich the introduction. The supplementary literature is as follows:

Zhang, B., Zhu, H., Dong, Z., & Wang, Q. (2022). Enhancement of bond performance of FRP bars with seawater coral aggregate concrete by utilizing ecoefficient slag-based alkali-activated materials. Journal of Composites for Construction, 26(1), 04021059.

Jahangir, H., & Esfahani, M. R. (2021). Experimental analysis on tensile strengthening properties of steel and glass fiber reinforced inorganic matrix composites. Scientia Iranica, 28(3), 1152-1166.

Wang, G., Wei, Y., Miao, K., Zheng, K., & Dong, F. (2022). Axial compressive behavior of seawater sea-sand coral aggregate concrete-filled circular FRP-steel composite tube columns. Construction and Building Materials, 315, 125737.

Point 2: What was the reason behind the selection of the fiber with specific dimensions/ aspect ratio. Need some clarity

Response 2: Thanks for your suggestions and questions. Too much or too little fiber content will have a negative impact on the mechanical properties of coral concrete to a certain extent. We consulted the relevant literature of fiber reinforced concrete, in order to facilitate the uniform distribution of fibers without agglomeration during pouring. And it can play a better role in bridging. Therefore, in this range, combined with the literature of relevant scholars, we finally selected 1kg·m-3, 2kg·m-3, 3kg·m-3 as the mass percentage of polypropylene fiber.

Point 3: Please include references for the standard tests.

Response 3: Thanks for your suggestions. We have added relevant explanations in the text and marked the corresponding parts in the text in red font.

The supplementary literature is as follows:

[24]China Building Materials Federation (2010) GB/T 17431.2-2010. General Administration of quality supervision, inspection and Quarantine of the people's Republic of China; China National Standardization Administration Committee.

[25]Ministry of construction of the people's Republic of China Industrial standard of the people's Republic of China JGJ 12-2006 / Technical Specification for lightweight aggregate concrete structures [M] China Construction Industry Press, 2006.

[26]GB500010-2010,Code for design concrete structures[S]. Beijing: China Architecture & Building Press, 2010. (in Chinese).

Point 4: There are no papers cited in the discussion section. Please include some papers to improve your discussion and also compare them with some other studies.

Response 4: Thanks for your suggestions. On the basis of the original text, the references of fiber reinforced concrete damage analysis, SEM micro analysis, coral concrete test, and the software website of equivalent energy method are supplemented. We have added relevant explanations in the text and marked the corresponding parts in the text in red font.

The supplementary literature is as follows:

[27]Liu C. *, Tang C., Shi B., Suo W., 2013. Automatic quantification of crack patterns by image processing. Computers and Geosciences, 57, 77-80. [doi: 10.1016/j.cageo.2013.04.008]

[28] Yang H ,  Liang D ,  Deng Z , et al. Effect of limestone powder in manufactured sand on the hydration products and microstructure of recycled aggregate concrete[J]. Construction and Building Materials, 2018, 188(NOV.10):1045-1049.

[29] Jidong Cui,“HLA:Hysteretic Loop Analysis Program", http:/www.idcui.com/?p=11828.

[30]Chen Zongping,Mo Linlin,Song Chunmei & Zhang Yaqi.(2021).Investigation on compression properties of seawater-sea sand concrete. ADVANCES IN CONCRETE CONSTRUCTION(2). doi:10.12989/ACC.2021.12.2.093.

Point 5: Overall, the paper has some interesting data, but needs major improvement in written English. There are many sentences forming errors that should be corrected before publication.

Response 5: Thanks for your suggestions. We will try to reduce the long sentences in the article as much as possible and check the grammar of the newly added content. We have added relevant explanations in the text and marked the corresponding parts in the text in red font. If you find any problems that need to be modified, please don't hesitate to contact us at any time: Tel: 1577808887; Email: [email protected].

Reviewer 3 Report

  1. Clear the contribution in introduction
  2. Clear the figures in results  
  3. Results need more discussions 

Author Response

Response to Reviewer 3 Comments

Dear reviewers:

First of all, thank you very much for taking the time to review our manuscript entitled “Experimental study on triaxial compressive mechanical properties of polypropylene fiber coral seawater concrete (materials-1728110)”. These comments are all valuable and very helpful for revising and improving our paper, as well as having important guiding significance to our researches.

We have studied comments carefully and have made correction which we hope meet with approval. Revised portions are marked in red in the paper. The related statements for these revisions are listed as follows.

Lastly, thanks very much for your fruitful work and patience.

Yours sincerely,

Hang Shi, Linlin Mo, Mingyan Pan, Leiguo Liu, Zongping Chen *.

May. 26, 2022

Point 1: Clear the contribution in introduction.

Response 1: Thanks for your suggestions. We have added relevant explanations in the text and marked the corresponding parts in the text in red font. The added contents are as follows: At present, the research results of coral aggregate concrete basically focus on the strength research level, or uniaxial compression test, and there is no in-depth discussion on the triaxial stress mechanism and stress performance of coral coarse aggregate concrete. This experiment was carried out for the first time in the world and has certain research significance.h as offshore engineering island reef construction.

Point 2: Clear the figures in results.

Response 2: Thanks for your suggestions. We integrated your opinions with the fourth editor and deleted the test conclusion drawn from quantitative analysis: The ductility of all-coral concrete specimens with fiber admixture V=3kg·m-3 was better. Some data results are retained to support the conclusion. For example, the growth multiple and the variation law of fiber content with the confining pressure value.

Point 3: Results need more discussions.

Response 3: Thanks for your suggestions. We have added relevant contents and marked in red font in the corresponding part of the text. Considering the sequence of this paper, the conclusions of specimen failure form, stress-strain curve, peak stress-strain elastic modulus analysis, ductility analysis, energy consumption analysis and damage analysis are relatively complete. However, the failure interface of the specimen is described in the macro and micro aspects of the article, so we add the relevant analysis obtained from the test to the conclusion. Therefore, the conclusion analysis of micro test is mainly added.

Reviewer 4 Report

In this manuscript, the mechanical properties of polypropylene fiber coral seawater concrete were investigated via triaxial compressive tests. The manuscript is well written and is of great importance in the related field of research. Some questions and suggestions are provided below to enrich the scientific level of the presented manuscript before its publication:

  • In usual, the amount of fibers express with volumetric percentage. Therefore, it is suggested to modify (1kg·m-3, 2kg·m-3, 3kg·m-3) as fiber presentation.
  • It is suggested to add results in quantity values at the end of the abstract.
  • The respectful author think can the manuscript be considered as an experimental study on lightweight structural concrete? If yes, it can be presented in the title and the keywords.
  • The introduction can be enrich by adding the following new published papers as reference:
    • Zhang, B., Zhu, H., Dong, Z., & Wang, Q. (2022). Enhancement of bond performance of FRP bars with seawater coral aggregate concrete by utilizing ecoefficient slag-based alkali-activated materials. Journal of Composites for Construction, 26(1), 04021059.
    • Jahangir, H., & Esfahani, M. R. (2021). Experimental analysis on tensile strengthening properties of steel and glass fiber reinforced inorganic matrix composites. Scientia Iranica28(3), 1152-1166.
    • Wang, G., Wei, Y., Miao, K., Zheng, K., & Dong, F. (2022). Axial compressive behavior of seawater sea-sand coral aggregate concrete-filled circular FRP-steel composite tube columns. Construction and Building Materials315, 125737.
  • The quality of presented six first figures can be improved.
  • Is there any possibility to utilize SEM analyses instead of presented Figures 8 and 9?
  • It would be great if the authors compare the proposed regression-based equations in Equations (1), (2), (3), and (6) with corresponding possible code-based equations.

Author Response

Response to Reviewer 4 Comments

Dear reviewers:

First of all, thank you very much for taking the time to review our manuscript entitled “Experimental study on triaxial compressive mechanical properties of polypropylene fiber coral seawater concrete (materials-1728110)”. These comments are all valuable and very helpful for revising and improving our paper, as well as having important guiding significance to our researches.

We have studied comments carefully and have made correction which we hope meet with approval. Revised portions are marked in red in the paper. The related statements for these revisions are listed as follows.

Lastly, thanks very much for your fruitful work and patience.

Yours sincerely,

Hang Shi, Linlin Mo, Mingyan Pan, Leiguo Liu, Zongping Chen *.

May. 26, 2022

Point 1: In usual, the amount of fibers express with volumetric percentage. Therefore, it is suggested to modify (1kg·m-3, 2kg·m-3, 3kg·m-3) as fiber presentation.

Response 1: Thanks for your suggestions and questions. Because the measurement of the volume proportion of fiber in concrete is not accurate, we choose the mass percentage, which is more convenient for scholars to carry out further experiments on the basis of this test. The mass percentage and volume percentage can be converted without affecting the expression of any data in this paper. However, if it is converted to volume percentage now, the data will be irregular, which may no longer be 1kg·m-3, 2kg·m-3, 3kg·m-3. Therefore, we hope that the editor will allow us to retain the expression of mass percentage.

Point 2: It is suggested to add results in quantity values at the end of the abstract.

Response 2: Thanks for your suggestions and questions. Due to the limitations of the test, the quantitative analysis can not represent a wide range of test results. Therefore, combined with the opinions of the first editor, we consider deleting the quantitative analysis, but we have modified and improved the conclusion.

Point 3: The respectful author think can the manuscript be considered as an experimental study on lightweight structural concrete? If yes, it can be presented in the title and the keywords.

Response 3: Thanks for your suggestions and questions. It can indeed be regarded as an experimental study of lightweight structural concrete. We have added relevant contents and marked in red font in the corresponding part of the text.

Point 4: There are no papers cited in the discussion section. Please include some papers to improve your discussion and also compare them with some other studies.

Response 4: Thanks for your suggestions. On the basis of the original text, the references of fiber reinforced concrete damage analysis, SEM micro analysis, coral concrete test, and the software website of equivalent energy method are supplemented. We have added relevant explanations in the text and marked the corresponding parts in the text in red font.

The supplementary literature is as follows:

[27]Liu C. *, Tang C., Shi B., Suo W., 2013. Automatic quantification of crack patterns by image processing. Computers and Geosciences, 57, 77-80. [doi: 10.1016/j.cageo.2013.04.008]

[28] Yang H ,  Liang D ,  Deng Z , et al. Effect of limestone powder

in manufactured sand on the hydration products and microstructure of recycled aggregate concrete[J]. Construction and Building Materials, 2018, 188(NOV.10):1045-1049.

[29] Jidong Cui,“HLA:Hysteretic Loop Analysis Program", http:/www.idcui.com/?p=11828.

[30]Chen Zongping,Mo Linlin,Song Chunmei & Zhang Yaqi.(2021).

Investigation on compression properties of seawater-sea sand concrete. ADVANCES IN CONCRETE CONSTRUCTION(2). doi:10.12989/ACC.2021.12.2.093.

Point 5: The introduction can be enrich by adding the following new published papers as reference:

Zhang, B., Zhu, H., Dong, Z., & Wang, Q. (2022). Enhancement of bond performance of FRP bars with seawater coral aggregate concrete by utilizing ecoefficient slag-based alkali-activated materials. Journal of Composites for Construction, 26(1), 04021059.

Jahangir, H., & Esfahani, M. R. (2021). Experimental analysis on tensile strengthening properties of steel and glass fiber reinforced inorganic matrix composites. Scientia Iranica, 28(3), 1152-1166.

Wang, G., Wei, Y., Miao, K., Zheng, K., & Dong, F. (2022). Axial compressive behavior of seawater sea-sand coral aggregate concrete-filled circular FRP-steel composite tube columns. Construction and Building Materials, 315, 125737.

Response 5: Thanks for your suggestions and questions. We have added relevant contents and marked in red font in the corresponding part of the text.

Point 6: The quality of presented six first figures can be improved.

Response 6: Thanks for your suggestions and questions. We have carefully revised the relevant graphs. By redrawing Figures 4, 5 and 6, and adding scales to figures 1, 2 and 3.

Point 7: Is there any possibility to utilize SEM analyses instead of presented Figures 8 and 9?

Response 7: Thanks for your suggestions and questions. Because SEM is equivalent to a microscope, only a small part can be observed in magnification, in order to better observe the damage of the interface in the macro. The later SEM test completely expressed the pictures of polypropylene fiber, coral aggregate, coral aggregate slurry interface transition zone, fiber aggregate interface transition zone and fiber slurry interface transition zone under different confining pressures. Therefore, it may be better to express it from both macro and micro perspectives.

Point 8: It would be great if the authors compare the proposed regression-based equations in Equations (1), (2), (3), and (6) with corresponding possible code-based equations.

Response 8: Thanks for your suggestions and questions. We fit the equation according to the classical equation based on code.

For example:

Equations (1): For formulas see Word Document         

Equations (2):  For formulas see Word Document

Equations (3):  For formulas see Word Document

Equations (6):  For formulas see Word Document

A new equation is derived by replacing the constants in the original literature with unknown quantities. Because the test material is different from the test curing method, the fitted equation will be different, and the comparative analysis may be of little significance

Round 2

Reviewer 2 Report

The Paper can be accepted since the authors have addressed my comments

Author Response

Response to Reviewer 2 Comments

Dear reviewers:

First of all, thank you very much for taking the time to review our manuscript entitled “Experimental study on triaxial compressive mechanical properties of polypropylene fiber coral seawater concrete (materials-1728110)”. These comments are all valuable and very helpful for revising and improving our paper, as well as having important guiding significance to our researches.

We have studied comments carefully and have made correction which we hope meet with approval. Revised portions are marked in red in the paper. The related statements for these revisions are listed as follows.

Lastly, thanks very much for your fruitful work and patience.

Yours sincerely,

Hang Shi, Linlin Mo, Mingyan Pan, Leiguo Liu, Zongping Chen *.

May. 28, 2022

Point 1: English language and style are fine/minor spell check required 

Response 1: Thanks for your suggestions. We have revised and polished some grammatical details of the original text. The revision marks are marked in blue on the basis of the first round of revision. We will enrich our research in the future.

Reviewer 4 Report

The revised manuscript applied all suggestions and answered to all questions. Therefore, can be accepted in the current version. 

The only point can be reminded is to check the reference numbers in the text and their consistency to the list of references at the end of the manuscript.

Author Response

Response to Reviewer 4 Comments

Dear reviewers:

First of all, thank you very much for taking the time to review our manuscript entitled “Experimental study on triaxial compressive mechanical properties of polypropylene fiber coral seawater concrete (materials-1728110)”. These comments are all valuable and very helpful for revising and improving our paper, as well as having important guiding significance to our researches.

We have studied comments carefully and have made correction which we hope meet with approval. Revised portions are marked in red in the paper. The related statements for these revisions are listed as follows.

Lastly, thanks very much for your fruitful work and patience.

Yours sincerely,

Hang Shi, Linlin Mo, Mingyan Pan, Leiguo Liu, Zongping Chen *.

May. 28, 2022

Point 1: The only point can be reminded is to check the reference numbers in the text and their consistency to the list of references at the end of the manuscript.

Response 1: Thanks for your suggestions. We checked the references of the full text, modified some formatting problems and marked them in blue

However, traditional gravel aggregates and freshwater resources on the island are extremely limited, and the transportation project from the mainland is costly and time-consuming [1].

[1]Da, B. Research on the preparation technology, durability and mechanical properties of concrete members of high strength coral aggregate seawater concrete. Ph.D. Dissertation, Nanjing University of Aeronautics and Astronautics, 2017.

Coral debris is light, porous, highly absorbent, loose in the structure, and rough on the surface, and belongs to natural lightweight aggregates [2].

[2]Chen, Z.P.; Zhou, J.; Chen, Y.L.; Yao, R.S. Experimental study on mechanical properties of coral coarse aggregate seawater concrete. Chin. J. Appl. Mechan. 2020, 37, 1999-2006.

However, coral inevitably has local cracks and infiltration as a porous material. Given the congenital disability that is prone to micro-cracks, if carbon steel is utilized, the corrosion resistance is poor in the harsh marine environment [3-4].

[3]Zhang, L.; Niu, D.T.; Wen, B.; Peng, G.; Sun, Z. Corrosion behavior of low alloy steel bars containing Cr and Al in coral concrete for ocean construction. Constr. Build. Mater. 2020, 258, 119564.

[4]Zhang, B., Zhu, H., Dong, Z., & Wang, Q. (2022). Enhancement of bond performance of FRP bars with seawater coral aggregate concrete by utilizing ecoefficient slag-based alkali-activated materials. Journal of Composites for Construction, 26(1), 04021059.

The addition of uniformly dispersed short fibers can significantly improve the defects of low tensile strength, poor toughness, and easy cracking of plain concrete [5], and microfibers can delay the development of macroscopic cracks [6].

  • Abdel-Kader, M.; Fouda, A. Effect of reinforcement on the response of concrete panels to impact of hard projectiles. J. Impact Eng.2014, 63, 1-17.

[6]Lawler, J.S.; Zampini, D.; Shah, S.P. Microfiber and macrofiber hybrid fiber-reinforced concrete. J. Mater. Civil Eng. 2005, 17, 595-604.

Polypropylene fibers thicken concrete, reduce concrete slump, improve crack resistance performance and inhibit early shrinkage cracks in concrete [7].

[7]Deng, M.L. Experimental research on the mechanical properties of PP fiber reinforced concrete and application on slope support project. Master’s Thesis, Chongqing University, 2013.

Given this, polypropylene fibers (PPF) with good toughening fracture-resisting capacity, corrosion resistance, odorless, non-toxic, high toughness, low price, and good chemical stability was chosen can be effectively dispersed in concrete [8].

[8]Bagherzadeh, R.; Sadeghi, A.H.; Latifi, M. Utilizing polypropylene fibers to improve physical and mechanical properties of concrete. Text. Res. J. 2012, 82, 88-96.

So far, scholars from various countries have achieved more fruitful results on the mechanical properties of coral concrete [9-11]. 

[9]Ke, X.; Fu, W.; Chen, Z. Mechanical properties of high-performance concrete under triaxial compression. Mag. Concrete Res. 2022, 74, 419-431.

[10]Ilg, M.; Plank, J. A novel kind of concrete superplasticizer based on lignite graft copolymers. Cement Concrete Res. 2016, 79, 123-130.

[11]Ye, P.; Chen, Y.; Chen, Z.; Xu, J.; Wu, H. Failure criteria and constitutive relationship of lightweight aggregate concrete under triaxial compression. Materials, 2022, 15, 507.

In 1974, Howdyshell P A [12], a U.S. Army Construction Engineering Laboratory scholar, published a research study on coral concrete and concluded that "it is feasible to formulate concrete using coarse coral aggregates, but the chloride salt content of coral aggregates should be controlled."

[12]Howdyshell, P.A. The Use of Coral as an Aggregate for Portland Cement Concrete Structures. Army Construction Engineering Research Laboratory, 1974.

Huang et al. showed through experimental comparison analysis that seawater increased the strength and elastic modulus of coral concrete compared to freshwater [13], and in 1991 Rick et al. [14] examined three coral concrete buildings on Bikini Island in the Pacific Ocean. They concluded that "the strength of coral concrete can meet the design requirements of engineering structures."

[13]Huang, Y.J.; Li, X.W.; Lu, Y.; Wang, H.C.; Wang, Q.; Sun, H.S.; Li, D.Y. Effect of mix component on the mechanical properties of coral concrete under axial compression. Constr. Build. Mater. 2019, 223, 736-754.

[14]Ehlert, R.A. Coral concrete at bikini atoll. Concrete Int. 1991, 13, 19-24.

 In 1996, the Indian scholar Arumugam R A et al. [15] conducted an extensive study and confirmed that coral concrete has a high early strength growth and a slow late strength growth.

[15]Arumugam, R.A.; Ramamurthy, K. Study of compressive strength characteristics of coral aggregate concrete. Mag. Concrete Res. 1996, 48, 141-148.

Daboo et al. [16] studied the mechanical properties of all-coral seawater concrete during uniaxial compression and showed that the uniaxially compressed specimens of all-coral seawater concrete still have high residual strength after damage and have some ductility.

[16]Da, B.; Yu, H.F.; Ma, H.Y.; Zhang, Y.D.; Yuan, Y.F.; Yu, Q.; Tan, Y.S.; Mi, Y.S. Experimental research on whole stress-strain curves of coral aggregate seawater concrete under uniaxial compression. J. Build. Struct. 2017, 38, 144-151.

Zhang et al. [17] used fiber-reinforced polymer to improve seawater coral aggregate concrete structures' bearing capacity and service performance in maritime environments.

[17]Jahangir, H., & Esfahani, M. R. (2021). Experimental analysis on tensile strengthening properties of steel and glass fiber reinforced inorganic matrix composites. Scientia Iranica, 28(3), 1152-1166.

 Wang et al. [18] conceived the unidirectional axial compression test of FRP steel composite circular tubular concrete column with circular saltwater, sea sand, and coral aggregate.

[18]Wang, G., Wei, Y., Miao, K., Zheng, K., & Dong, F. (2022). Axial compressive behavior of seawater sea-sand coral aggregate concrete-filled circular FRP-steel composite tube columns. Construction and Building Materials, 315, 125737.

The experimental consequences of Arefi et al. [19] showed that coarse polypropylene fibers improved the ultimate load-bearing capacity and energy absorption capacity of concrete.

[19]Arefi MR.; Mollaahmadi E. An experimental investigation into the effect of polypropylene fibers on mechanical properties of concrete. 2012, 745-753.

Bagherzadeh et al. [8] concluded that the PPF with a length of 19mm improved the toughness index of concrete.

[8]Bagherzadeh, R.; Sadeghi, A.H.; Latifi, M. Utilizing polypropylene fibers to improve physical and mechanical properties of concrete. Text. Res. J. 2012, 82, 88-96.

Wang et al. [20] pointed out that polypropylene fibers make concrete with lower elastic modulus, reduced shrinkage, and more robust deformation performance with the same water-cement ratio and the same slump conditions.

[20]Wang, Z.Z. Experiment study on the properties of Polypropylene fiber concrete. Master’s Thesis, Zhejiang University, 2004.

 Xu et al. [21] concluded from uniaxial cyclic loading tests that incorporating polypropylene fibers significantly improved the mechanical compressive cyclic behavior of concrete, increasing its compression toughness, and final peak ductility hysteretic energy capacity, and reducing its rigidity degradation and stress deterioration.

[21]Xu, L.H.; Huang, B.; Li, B.; Chi, Y.; Li, C.N.; Shi, Y.C. Study on the stress-strain relation of polypropylene fiber reinforced concrete under cyclic compression. China Civil Eng. J. 2019, 52, 1-12.

The subtle blending rate of synthetic fibers can play a significant role in concrete crack resistance and toughening [22]. Too high fiber blending significantly reduces its work performance [23].

[22]Shen, R.X. Mechanism of low synthetic fiber in concrete. In Academician Wu Zhongwei has been engaged in science and education work for 60 years, Beijing, China, 30 June 2004, 12-15.

[23]Wang, L.; Shen, N.; Yu, D.P. Strengthening mechanism and microstructures of fiber reinforced coral concrete. In Proceedings of the Institution of Civil Engineers - Structures and Buildings, 2021.

"Light Aggregate and its test method" (GB/T 17431.2-2010)[24]

[24]China Building Materials Federation (2010) GB/T 17431.2-2010. General Administration of quality supervision, inspection and Quarantine of the people's Republic of China; China National Standardization Administration Committee.

"light aggregate concrete structure technical regulations" (JGJ 12-2006)[25]

[25]Ministry of construction of the people's Republic of China Industrial standard of the people's Republic of China JGJ 12-2006 / Technical Specification for lightweight aggregate concrete structures [M] China Construction Industry Press, 2006.

Mechanical Properties Test Methods for Ordinary Concrete (GB/T 50081-2019)[26] 

[26]GB500010-2010,Code for design concrete structuresï¼»Sï¼½. Beijing: China Architecture & Building Press, 2010. (in Chinese).

Figure 8 shows the destructive form of some specimens. Using PCAS software to map the specimen fracture, it can be seen that the destructive form of the specimen changes from axial splitting failure to diagonal shear failure and finally to transverse shear failure[27].

[27]Liu C. *, Tang C., Shi B., Suo W., 2013. Automatic quantification of crack patterns by image processing. Computers and Geosciences, 57, 77-80. doi: 10.1016/j.cageo.2013.04.008

Coral aggregates have a more pronounced apparent density and internal friction angle. It helps to increase the interfacial bond between the aggregate and the hardened mortar. To some extent, it improves the interfacial strength of concrete [28].

[28]Yang H , Liang D , Deng Z , et al. Effect of limestone powder in manufactured sand on the hydration products and microstructure of recycled aggregate concrete[J]. Construction and Building Materials, 2018, 188(NOV.10):1045-1049.

Figure.27 Schematic diagram of energy consumption calculation[29]

[29]Jidong Cui,“HLA:Hysteretic Loop Analysis Program", http:/www.idcui.com/?p=11828.

Damage process analysis[30].

[30]Chen Zongping,Mo Linlin,Song Chunmei & Zhang Yaqi.(2021).Investigation on compression properties of seawater-sea sand concrete. ADVANCES IN CONCRETE CONSTRUCTION(2). doi:10.12989/ACC.2021.12.2.093.
